# Anthropogenic Aerosol effects on Tropospheric Circulation and Sea Surface Temperature (1980-2020): Separating the role of Zonally Asymmetric Forcings

Chenrui Diao[1], Yangyang Xu[1,*], Shang-Ping Xie[2]

5   [1] Department of Atmospheric Sciences, Texas A&M University, College Station, Texas 77843, USA

[2] Scripps Institute of Oceanography, University of California, San Diego, La Jolla, CA 92093, USA

*Correspondence to*: Yangyang Xu (yangyang.xu@tamu.edu)

**Abstract.** Anthropogenic Aerosols (AA) induce global and regional tropospheric circulation adjustments due to the radiative energy perturbations. The overall cooling effects of AA, which mask a portion of global warming, have been the subject of many studies but still have large uncertainty. The interhemispheric contrast in AA forcing has also been demonstrated to induce a major shift in atmospheric circulation. However, the zonal redistribution of AA emissions since the 20th century, with a notable decline in the Western Hemisphere (North America and Europe) and a continuous increase in the Eastern Hemisphere (South Asia and East Asia), received less attention.

Here we utilize four sets of single-model initial-condition large-ensemble simulations with various combinations of external forcings to quantify the radiative and circulation responses due to spatial redistribution of AA forcing during 1980-2020. In particular, we focus on the distinct climate responses due to Fossil-Fuel (FF) related aerosols emitted from the Western Hemisphere (WH) versus the Eastern Hemisphere (EH).

The zonal (West to East) redistribution of FF aerosol emission since the 1980s leads to a weakening negative radiative forcing over the WH mid-to-high latitudes and an enhancing negative radiative forcing over the EH at lower latitudes. Overall, the FF aerosol leads to a northward shift of Hadley Cell and an equatorward shift of the Northern hemisphere (NH) jet stream. Two sets of regional FF simulations (Fix_EastFF1920 and Fix_WestFF1920) are performed to separate the roles of zonally asymmetric aerosol forcings. We find that the WH aerosol forcing, located in extratropic, dominates the northward shift of Hadley Cell by inducing an interhemispheric imbalance in radiative forcing. On the other hand, the EH aerosol forcing, located closer to the tropics, dominates the equatorward shift of the NH jet stream. The consistent relationship between the jet stream shift and the Top-of-Atmosphere net solar flux (FSNTOA) gradient suggests that the latter serves as a rule-of-thumb guidance for the expected shift of the NH jet stream.

The surface effect of EH aerosol forcing (mainly from low-to-mid latitudes) is confined more locally and only induces weak warming over the northeastern Pacific and North Atlantic. In contrast, the WH aerosol reduction leads to a large-scale warming over NH mid-to-high latitudes that largely offsets the cooling over the northeastern Pacific due to EH aerosols.

The simulated competing roles of regional aerosol forcings in driving atmospheric circulation and surface temperature responses during the recent decades highlight the importance of considering zonally asymmetric forcings (West to East) and also their meridional locations within the NH (tropical vs. extratropical).

# 1 Introduction

The external forcings due to anthropogenic activities and internal variabilities originating from the ocean-atmosphere system together determine climate change at decadal time scales (Kirtman et al., 2013; Meehl et al., 2013). Since the Industrial Revolution, the increasing GHG emissions have been shown to be the leading cause of global warming of about 1.1 ℃ (as of the late 2010s; IPCC, 2018). On the other hand, the internal variation of the climate, which fluctuates at time scales ranging from years to decades, modulate the paces of global warming at a shorter decal to a multi decadal time scale (Dai et al., 2015; Xie and Kosaka, 2017; Dong and McPahden, 2017), also with regional implications such as sea ice retreat (Ding et al., 2019).

In addition to GHG forcing and internal variability, another major confounding factor affecting global climate change at decadal scales is anthropogenic aerosol forcing. Despite decades of research into this subject, quantitative understandings of the regional climate effects of Anthropogenic Aerosols (AA) remain highly uncertain. There is still limited understanding of the physical mechanisms governing the strength of AA radiative forcing, for example, due to complex aerosol-cloud interaction (Fiedler et al., 2017; Bender, 2020), the brownness of organic aerosols (Bahadur et al., 2012; Jacobson, 2012; Kodros et al., 2015), and surface albedo changes due to black carbon aerosols (Xu et al., 2016; Liu et al., 2020).

Additionally, there are at least two more reasons why a robust attribution of past climate change to AA is difficult: uneven spatial distributions and fast temporal evolutions of emission/concentration/forcing. Unlike GHGs, the lifetimes of aerosols are as short as days, and thus the spatial distribution of aerosol concentration and its forcing is highly heterogeneous, which may perturb regional climate differently compared to the well-mixed GHG (Ming and Ramaswamy, 2011; Shindell et al., 2015). Lin et al. (2018) analyzed the relationship between aerosol and precipitation extremes and showed that precipitation extremes are more sensitive to aerosols than GHGs, consistent with Salzmann (2016) which examined global mean precipitation. Also, the relatively shorter lifetime means that AA concentrations respond to local emission changes quickly. Indeed, global sulfate aerosol concentration has declined following strengthened emission control measures in the developed nations in the West (Klimont et al., 2013), in contrast to the monotonic increase in GHG concentration since the industrial revolution.

However, despite the differences between GHGs and aerosols, other studies found similar climate responses to GHGs and aerosols. Xie et al. (2013) found that the 20th century regional temperature and precipitation are similar in response to GHGs and the more spatially heterogeneous aerosol forcing. Song et al. (2021) showed that increasing GHGs and decreasing aerosols in the recent decades both delay rainfall by inducing a moister atmosphere. Both the differences and similarities between GHGs- and aerosol-induced climate responses indicate the complexity and importance of the temporal and spatial distribution of AA forcings.

Because of the unique temporal and spatial features of AA, some have argued that the aerosol forcing can induce an external-forced "decadal variability", which can then be imposed onto the natural variabilities, further confounding a robust attribution of observed changes at a shorter time scale. For example, several recent studies suggested the reduction in aerosol emission over Europe contributes to the Atlantic Multidecadal Variability, which is to a large part attributed to internal oceanic processes (Booth et al., 2012; Bellomo et al., 2018; Hua et al., 2019; Watanabe and Tatebe, 2019). The Atlantic Meridional Overturning Circulation (AMOC) is also argued by recent studies to be induced by AA forcing (Hassan et al., 2020; Menary et al., 2020), though with large uncertainties. Some studies focused on aerosol effects on the Pacific decadal to multidecadal variations, arguing that aerosol forcings can induce Pacific decadal variation (Allen et al., 2014; Dong et al., 2014; Hua et al., 2018), but the relative contribution of external forcing and internal variability remains unclear.

Given the rapid temporal evolution of global aerosol emission, as well as the regional redistribution of dominant aerosol emission regions from the West to East, the question of how AA affects the regional climates needs further investment. Many previous studies examined the aerosol geographical distribution effect on circulation, forcing and temperature (Chemke et al., 2018; Shen et al., 2018). The recent aerosol unmasking in the West can have profound implications on regional climate (Samset et al., 2018; Zhao et al., 2019; Wang et al., 2020b), Arctic sea ice (Krishnan et al., 2020), etc. Wang et al. (2015) demonstrated that the redistribution of aerosol from west to east induces a southward shift of circulation systems and the weakening of tropical circulation. Wang et al. (2020a) showed a large shift in South Hadley circulation due to AA in the 20th century. Recently, Wang et al. (2020b) demonstrated that the reduced aerosol emission over Europe suppresses the Eurasia wintertime extremes.

Recently, a few studies attempted to separate the potential competing effects of regional aerosol forcings. Persad and Caldeira (2018) demonstrated diverse temperature responses due to regional aerosol forcings based on a set of idealized model simulations with identical emission placed at different continents. Kang et al. (2021) examined the climate responses to the zonal shift (West to East) of aerosol forcing but did not consider the meridional difference of West and East aerosol forcings. Understanding the climate response to spatial (zonal and meridional) redistribution of aerosol forcing is the main motivation of the present study. We aim to separate and compare the potential competing roles of increasing aerosol forcing from the Eastern Hemisphere (EH) and decreasing aerosol forcing from the Western Hemisphere (WH) based on a suite of large ensemble simulations with temporally evolving aerosol forcings.

An improved understanding can help shed light on other relevant problems on regional forcings, such as land-use changes (e.g., deforestation over Amazon vs. Africa), volcanic eruption (Verma et al., 2019), geoengineering solutions such as stratospheric or tropospheric aerosol injection conducted over different locations, and the potential contrast of China and India's future emission trajectories in future decades (Samset et al., 2019; Wang et al., 2021). The FF-related aerosols are

projected to further decrease in future decades (Andreae et al., 2005; Zheng et al., 2020), even for Asian regions, with more strict air quality measures in developing nations. The future decline of FF aerosol will lead to further unmasking and warming in addition to GHG-induced global warming (Xu and Xie, 2015; Lelieveld et al., 2019; Allen et al., 2020; Wang et al., 2020a) and have consequences for heat extremes (Zhao et al., 2019; Xu et al., 2020) and humidity and precipitation (Song et al., 2021).


This study leveraged a recently available large ensemble simulation using the fully coupled global climate model and conducted additional "regional" single forcing experiments to assess the aerosol impact on global climate change in the past few decades (1980-2020). The present study focuses on the zonal (WH to EH) asymmetry of aerosol forcing within the Northern Hemisphere. We aim to detail how the upward EH AA emission trend and the downward WH AA trend competes

to affect tropical and mid-latitude circulation, and simultaneously, affect the North Pacific surface climate which may have played a role in determining the observed Pacific decadal variations.

The structure of this paper is the following. In Sect. 2, we provide the details of the climate model, published simulation, and our new model experiment. In Sect. 3, we present simulated responses on the global and regional radiation budget (Sect.

3.1), air temperature (Sect. 3.2), and NH tropospheric circulation (Sect. 3.3) with a focus on separating the role of WH and EH FF forcing that have clear zonal asymmetry. The importance of the latitudinal distribution of AA forcing in driving North Pacific temperature change is highlighted in Sect. 3.4. In Sect. 4, we summarize our findings and suggest scientific questions for future research.

**2 Methods**

**2.1 Climate model**

The climate model used in this study is the Community Earth System Model 1 (CESM1). CESM1 is a fully coupled model developed by NCAR and community scientists (Hurrell et al., 2014) and is one of the models participating in the Coupled Model Intercomparison Project Phase 5 (CMIP5) (Meehl et al., 2016). The CESM1 has been extensively applied in a variety

of climate studies including the ones focusing on external forcing and internal variability (e.g., Swart et al., 2015; Xu et al., 2015; Kay et al., 2015; Ding et al., 2019). Studies utilizing CMIP5 multi-model comparison (e.g., Samset et al., 2016; Smith et al., 2016; Lin et al., 2018) also demonstrated its capability for attribution studies on human-induced regional climate change.

Relevant to the aerosol effect focused on this study, a scheme of the three-mode aerosol model (MAM3) - Aitken, accumulation, and coarse modes (Liu et al., 2012), is used by default in CESM1 (CAM5). Aerosol concentration (including sulfates, black carbons, organic carbons) in CESM1 (CAM5) is calculated online from the historical (up to 2005) and future (RCP8.5 thereafter) emission scenarios. The cloud physics scheme allows ice supersaturation and features activation of aerosols to form cloud droplets and ice crystals and thus enables simulations of aerosol indirect effects (Morrison and

Gettlemen, 2008), which was missing in the model's predecessors.

The simulations used in this study are based on a model version of nominal 1° horizontal resolution (0.9° X 1.25°) and 30 vertical levels. All simulation outputs analyzed in this study are monthly data.

**2.2 Existing simulations**

Our study relies on two published large ensemble datasets using CESM1 (CAM5):
a. CESM1 Large Ensemble Project (CESM1-LENS; Kay et al., 2015).
b. CESM1 "Single Forcing" Large Ensemble Project (Deser et al., 2020).

The CESM1-LENS includes a 40-member ensemble of fully coupled simulations for the period of 1920-2100 with the same historical radiative forcing up to 2005 and the RCP8.5 scenario thereafter (Kay et al., 2015). Each ensemble member starts from the same simulation restart file in 1920 but with slightly different air temperatures perturbed at the level of round-off error. In this paper, we use "ALL" (i.e., all forcing considered) to represent this large ensemble. One advantage of having a large ensemble simulation is that we can separate climate responses to external forcings from internal variabilities by

ensemble averaging. Thus, all results in this study are based on the ensemble average.

The CESM1 "Single Forcing" Large Ensemble uses the same model setup of the CESM1-LENS but with individual external forcing fixed at the 1920 level while keeping all other external forcing evolving with time into the 21st century. The "Single Forcing" Large Ensemble includes four sets of ensembles with different single forcings fixed: (1) industrial aerosols

(XAERindus, 20 members, 1920-2080), (2) biomass burning aerosols (XAERbmb, 15 members, 1920-2029), (3) greenhouse gases (XGHGs, 20 members, 1920-2080), and (4) land-use/land-cover (XLULC, 5 members, 1920-2029). Here in this study, we only used the first two ensembles. We changed the notation of the two ensembles to "Fix_FF1920" ("FF" stands for Fossil Fuel) and Fix_BB1920 ("BB" stands for Biomass Burning) because, in the emission inventory dataset, energy/transportation sector-related emission is also fixed, rather than the industrial activities only as the original notation

implies. We also emphasize the timing (the year 1920) of leveling emission here because anthropogenic aerosol emissions in these simulations are not removed entirely but rather stay at a relatively low level (blue lines in Fig. 1 a and b).

By subtracting the "Fix_FF1920" or "Fix_BB1920" ensemble average results from the "ALL" ensemble average, we can obtain climate responses to the Fossil-Fuel-related aerosol forcing (FF) or Biomass-Burning aerosol forcing (BB). Note that other sets of single forcing simulations (e.g., historicalMisc cases in CMIP5 (Taylor et al., 2012), and "hist-aer" cases in CMIP6 (Gillett et al., 2016)) simulate only historical aerosol evolution with all other forcings fixed at pre-industrial state. The fix-aerosol method adopted here (as in Deser et al., 2020, but also in earlier studies such as Xu et al., 2015) serves to estimate the aerosol effects with all other external forcings (such as GHGs) evolving in the background, arguably an advantage in experimental design to assess the actual impact of single forcing.

One potential issue of using the fixed single forcing approach is that we have assumed additivity when differencing "ALL" and fixed single forcing cases (i.e., Fix_FF1920 or Fix_BB1920). The additivity assumption is examined in several recent studies focusing on the nonlinear interaction between aerosols and GHGs (Deng et al., 2019) and for various climate variables, in particular for extreme precipitation (Lin et al., 2018).

## 2.3 New simulation

The existing two sets of single forcing large ensemble simulations (Fix_FF1920 and Fix_BB1920) enables a robust separation of aerosol-induced responses and a comparison of the role of FF and BB forcing. However, FF forcing features a strong zonal asymmetry starting from the 1980s (blue solid vs. dashed lines in Fig. 1 c, d), which continuously increases over the EH (dashed lines) and decreases over the WH (solid lines). The sharp contrast and competition between WH and EH in FF emission and forcing trend (driven by different air pollution policy) brings extra complexity to our attribution.

To gain further insights on the role of regional forcing (East vs. West), we conducted two additional sets of "regional" single forcing large ensemble simulations (10 realizations for each case) by branching from the existing run of Fix_FF1920. In the experiment of Fixed *Eastern* Fossil Fuel simulation (Fix_EastFF1920), we use the same initialization protocol as Fix_FF1920, but only fix the aerosols over the EH box (0–80 ºN, 60–150 ºE; shown as the blue dashed box in Fig. 2), where FF aerosol emissions over other regions are allowed to evolve, including the decline over North America and Europe (shown as the blue solid box in Fig. 2). The experiment of Fixed *Western* Fossil Fuel simulation (Fix_WestFF1920) is similar to the setup of Fix_EastFF1920 case except that we fix the aerosols over WH box (20º–80ºN, 130º–10ºW, and 30º–80ºN, 10ºW–40ºE; shown as the blue dashed box in Fig. 2). We run the two sets of simulations from 1920 through 1980 for one realization, and then expand the ensemble size to be 10 for 1980–2020. A small random perturbation of surface temperature is applied to each realization to generate ensemble spreads.

In Fix_EastFF1920, except for the increasing negative forcing in the lower latitudes of East Asia, Siberia shows a slight weakening of the negative forcing (positive anomaly of radiative forcing) due to the extension of WH aerosol reduction. However, the extended positive radiative forcing anomaly is considerably weak compared to the negative forcing and is largely constrained in the small emission domain. Therefore, the difference between ALL and Fix_EastFF1920 can be safely used to represent the climate in response to the dominant role of the negative radiative forcing from lower latitudes of Asia.

Similar to how we obtain FF response, we subtract the ensemble average results of Fix_EastFF1920 and Fix_WestFF1920 from the ALL respectively to obtain climate responses to regional Fossil-Fuel-related aerosol forcings (EastFF and WestFF). An additivity test is conducted to evaluate whether the summation of EastFF and WestFF can roughly reproduce FF. The SO4 column burden (BURDENSO4) and surface temperature in response to FF and EastFF + WestFF (SUM hereafter) are shown in Fig. 3. The FF-induced SO4 column burden resembles the sum of the SO4 burden from SUM. The surface temperature responses are also very similar between FF and SUM, except for the central Pacific and part of the Arctic region. The warm bias over the central Pacific in SUM is possibly associated with forcings outside the two focused regions (EH box and WH box in Fig. 2), or it is due to the residual effect of internal variability even after ensemble average due to limited ensemble sizes. Overall, the sum of two sets of regional fixed single forcing experiments well represents the major patterns of FF aerosol induced response, and thus the two new sets of simulations here are capable of separating the East versus West aerosol forcings.

Note that we did not conduct the analogous simulation for BB because, for NH, significant BB emission and forcing trends during 1980-2020 are only over the EH (green dashed lines in Fig. 1c and d), specifically from Northeast Asia. Thus, the existing Fix_BB1920 simulation already captures the regional contribution from the EH.

## 3 Results

### 3.1 Zonal asymmetry of anthropogenic aerosol forcing in the recent decades

Figure 1 shows the global and regional emissions of two major types of AA (sulfur aerosols and organic carbon). Globally, it is clear that the AA emission started to decline since the late 20th century (shaded area of 1980-2020 as the focused period of this study), but the decrease in aerosols mainly comes from developed countries in North America and Europe (solid lines as Western Hemisphere (WH) in Fig. 1c, d) while the developing countries in Asia (e.g., China and India) are still in the phase of increasing aerosol emission (dashed lines as Eastern Hemisphere (EH) in Fig. 1c, d).

The 1st row of Fig. 2 depicts the 40-year linear trend of the SO4 column burden between 1980 and 2020. SO4 trend, as the dominant cooling aerosol produced by FF, shows a clear heterogeneous pattern in NH, with a decrease over North America and Europe (shown as the blue solid box) and a strong increase over China and India (shown as the lower part of the blue dashed box). Note that the decrease in SO4 column burden also occurs over the mid-to-high latitudes of Asia, though with weaker trends compared to Europe and North America.

Unlike Sulfate, another major cooling aerosol species, primary organic matter (POM) burden shows different distributions (2nd row of Fig. 2). FF-related POM is similar to that of SO4 burden but with much weaker negative trends over North America and also a weaker positive trend over China, compared to SO4. In contrast, BB-related POM features a much stronger increasing trend over northeastern Asia (40 ºN–70 ºN, 70 ºE–150 ºE), and a slight decrease over rain forests of Amazon and Congo. Combining FF and BB, the significant increasing trend of POM occurs over Asia at both low latitudes and high latitudes, while a relatively weaker decline trend can be found over Europe, Africa, and South America, again constituting a west-east zonal asymmetry. The Secondary Organic Aerosol (SOA) burden resembles the SO4 burden in both FF and BB cases but with weaker trends.

Aerosols with heating effects (such as Black Carbon; not shown) resemble the spatial pattern of SO4 burden shown in Fig. 2. However, the overall aerosols effect is dominated by cooling aerosols such as SO4. Thus, in this study, we only focus on the total cooling effect of aerosols, without separating the warming and cooling competition as done in several earlier studies (Xu and Xie, 2015; Lin et al., 2016; Wang et al., 2017).

Looking at aerosol mass burden only is insufficient to establish connections between the radiative forcing response and aerosol emissions because different aerosol species could have different radiative forcing efficiency. Thus, we further show Aerosol Optical Depth due to anthropogenic aerosol emission (AOD_AA) in the bottom panels of Fig. 2. To remove AOD induced by natural aerosols such as dust and sea salt, we derive AOD_AA following Eq. (1), in which, the AODVIS is the total AOD at the 550nm band, and the AODDUST 1-3 represent the dust AOD with different sizes. The background AOD (bkg_AOD) is the 100-year climatology of (AODVIS - AODDUST(1-3)) in the CESM1 pre-industrial control run, which is dominated by sea salt.

$$AOD\_AA = AODVIS - AODDUST(1-3) - bkg\_AOD \,, \tag{1}$$

As expected, the AOD_AA trend (third row in Fig. 2) in response to FF resembles the SO4 burden (first row in Fig. 2), while AOD_AA in response to BB is in close agreement with the POM. Both FF and BB AOD trends feature zonal asymmetry across the Pacific Ocean, with differences in terms of latitudinal distribution (increase at lower latitudes versus decrease at higher latitudes in FF. The implications of these spatial contrasts on climatic responses will be further discussed in the next section.

It is clear that BB shows a simple distribution without zonal competition, where a significant increase occurs over northeastern Asia. Therefore, our following discussion will only focus on the FF responses, which show significant zonal differences. Based on the released simulation and the new regional-FF simulations, we are able to separate the climate responses to aerosol increase over EH and aerosol reduction over WH.

In line with the zonal asymmetry of AOD_AA trends, simulated solar radiation flux also has significant zonal contrast due to aerosols' direct and indirect climate effects. The first row of Fig. 3 shows the Surface Downward Solar radiation (FSDS), broadly consistent with the patterns of the AOD_AA trend (third row of Fig. 2). Note the opposite colors, though, because a decline in AOD_AA leads to an increase in FSDS. The global surface radiative forcing shows an overall positive trend in response to the decrease in global sulfate emission, but with significant spatial heterogeneity due to the opposite regional emission trends. An increase in FSDS occurs over North America, Europe, and the northern part of Asia, consistent with reducing aerosol forcings over these regions. In contrast, the increasing aerosol emission over east Asia induces a substantial decrease in FSDS. The Net Solar Radiation at the Top-Of-Atmosphere (FSNTOA; the second row of Fig. 3), as the main metric for aerosol forcing, is also consistent with FSDS patterns, but shows more obvious responses over the ocean. Both FSDS and FSNTOA show significant trends over not only the emission domain but also over extended regions into the ocean surface. In response to WestFF, the north Atlantic region shows strong increases in solar radiations, which is consistent with the significant decrease in cloud droplet number concentration (CDNC, third row of Fig. 3). However, the cloud fractions (fourth row of Fig. 3) show very weak changes over the north Atlantic, which indicates the critical role of the aerosol first indirect effect over the north Atlantic (Penner et al. 2001). The reduction of aerosol emission over North America also leads to smaller CDNC over the North Pacific in the WestFF case, further contributing to a positive radiative forcing anomaly. In contrast, in response to EastFF, CDNC shows a significant increase over the subtropical Pacific in the Northern Hemisphere, which is consistent with the weak increase in SO4 burden over this region. The larger CDNC increases cloud albedo and amplifies the negative radiative forcing. The negative radiative forcing over the subtropical Pacific is evident in the EastFF case, but weaker in the (total) FF case due to the offset by the decreasing aerosols from North America.

The cloud fractions over this region also show an increasing trend in the FF and EastFF but fail to pass the significance test in the FF case. Surprisingly, the eastern subtropical Pacific in the South Hemisphere also shows significant changes in TOA solar radiation and the cloud fraction, without much aerosol changes. This may possibly be explained by the slow response of sea surface temperature (SST) to the aerosol forcing, where the cloud fraction is affected by the climate adjustment due to SST or circulation changes (Xu and Xie, 2015; Wang et al., 2016; Dong et al., 2019; Kang et al., 2021). The slow responses of cloud fraction to aerosol forcing could also occur near the emission regions where SST changes more significantly; however, as discussed above, the simulated radiation changes over and near the emission regions are highly consistent with the changes in CDNC, indicating a dominant role of indirect aerosol forcing through microphysics perturbation. Here we

mainly focused on the overall circulation changes in response to regional aerosol forcings using a fully coupled climate model, therefore a clear separation of the slow and fast responses of clouds and climate to aerosol forcing is beyond the scope of this study.

The North Pacific region, a focused region of this study, shows complex competition of the two emission sources, where WestFF induces a significant decrease in cloud droplet concentration (along with increasing FSNTOA) northward of 30 ºN. In contrast, EastFF leads to an opposite trend at 30 ºN and south. One may expect an increase in FF aerosol over Asia would lead to a negative forcing trend over the North Pacific, as is claimed in previous studies, but actually, the simulated negative

trends are confined to lower latitude regions (30 ºN and south). The two sets of regional forcing simulations reveal clearly that the decline of FF aerosol over the WH mid-latitudes induces the positive radiative forcing anomaly at mid-high latitudes of the North Pacific, producing a weak positive FSNTOA trend. This demonstrated East-West competition is a focal point of our following analysis. In the subsequent sections, we will discuss the possible mechanisms in terms of temperature and circulation changes.

### 3.2 Simulated responses in the hemispheric average of surface air temperature

In Sect. 3.1, we demonstrate the distinct East-West pattern of the aerosol emission changes and its radiative effect. This section analyzes the simulated response in the hemispheric average of Surface Air Temperature (SAT).

Figure 5a shows the Northern Hemisphere (NH) mean SAT from the two large ensemble simulations. Without the FF aerosol emission in Fix_FF1920 simulation (blue line in Fig. 5a), NH-mean SAT is significantly warmer than the air temperature in the ALL simulation (black line in Fig. 5a). Large volcanic eruptions (Four major ones are shown as vertical dashed lines in Fig. 5a) also strongly affect the NH-mean SAT by causing abrupt cooling of about 0.1 to 0.3 K episodically but the cooling effects quickly recover in a few years, which means it hardly affects the multidecadal climate trend. The

global-mean SAT evolutions resemble the NH result but with a weaker magnitude. The stronger response over NH is reasonable because most of the emission sources (and land regions) are located at NH and aerosol burden and radiative forcing (Fig. 2 and 3) are regionally concentrated.

Fig. 5b shows the climate response to FF by calculating the difference between ALL and Fix_FF1920 simulation. It indicates

that the mid-20th-century aerosol cooling effect is dominated by FF aerosol (Meehl et al., 2004; Diao and Xu., in review) with a cooling trend of 0.16 K/decade over NH. Since the 1980s, as the aerosol emission started to decline over WH, the NH-mean SAT response to FF aerosol has shifted to a slightly warming trend by about 0.07 K/decade.

Because of the distinct East-West aerosol forcing asymmetry, we further examine how the aerosol emissions would
influence regional SAT differently. Fig. 5c–d shows the temperature responses over the Eastern (80 ºE–140 ºW) and the
Western (90 ºW–30 ºE) portion of NH separately. The domains are shown as the red boxes in Fig. 4 but here we only have
the NH portion to be considered to be consistent with Fig. 5a and b. The Western-NH temperature in response to FF (Fig.
5d) is largely following local emission evolution (Fig. 1), with a cooling along with the increasing emission before the 1980s
and warming along with emission reduction afterward.

Notably, the Eastern-NH responses to FF are counter-intuitive: There is a warming trend of 0.06 K/decade (same as
Western-NH) after the 1980s (Fig. 5c) even with continuously increasing aerosol emission over this region, suggesting that
Eastern-NH is sensitive to the remote influence of WH aerosols. Indeed, the Eastern-NH cooling response is even larger than
the Western-NH response during the previous cooling period (1940-1980; -0.17 K/decade versus -0.11 K/decade). The
apparent contradiction of the EH warming in response to FF (Fig. 5c) and the local negative FF forcing (3rd row of Fig. 2)
bears important implications on tropospheric temperature and circulation changes (to be further explored in Sect. 3.4). Here
we argue that the larger remote response of EH temperature to WestFF is due to the latitudinal difference in aerosol emission
location: WH emission changes mainly occur over mid-to-high latitudes (30–60ºN; the first row of Fig. 2), while the EH
emission changes are mainly located over low-to-mid latitudes (southward of 40ºN, the first row of Fig. 2). As a result,
during 1980-2020, there is only a weak regional cooling over EH, due to local FF aerosols and is confined over low latitudes
of EH; but the WH decline of emission (positive radiative forcing anomaly) dominates the mid-to-high latitudes SAT
change, including North Pacific. Detailed analysis of this intriguing latitudinal contrast in forcing will be provided in Sect.
3.4 for regional SAT over North Pacific.

**3.3 Tropospheric responses**

Because of the complex zonal and meridional differences in aerosol emission during 1980-2020 (Sect. 3.1), and the
competition between EastFF and WestFF in changing NH air temperature (Sect. 3.2), tropospheric circulation responses
could also be distinct over different regions. In this subsection, we discuss the global and regional tropospheric circulation
responses due to the evolving anthropogenic aerosol emission, which have a major implication on mid-latitude climate (Xu
and Xie, 2015; Mann et al., 2017; Wang et al., 2020b).

Previous studies have explored the tropospheric circulation responses to inter-hemispheric (meridional) forcing gradient due
to anthropogenic aerosols – more reflecting aerosols over NH compared to SH will lead to an equatorward shift of NH
Hadley Cell and NH westerly wind (e.g., Hwang et al., 2013; Hilgenbrink et al., 2018). Meanwhile, recent studies also put
effort into how the west-east contrast effects of aerosol induce the circulation changes (Wang et al., 2015; Kang et al., 2021).

However, from 1980 to 2020, NH anthropogenic aerosol forcing (Sect. 3.1) is highly heterogeneous, with both zonal and latitudinal contrasts (Fig. 4), further compounding the forcing-response relationship (Shindell and Faluvegi, 2009; Persad and Caldeira, 2018). Next, we will analyze the aerosol-induced tropospheric responses (in terms of zonal average) both globally and regionally, for the EH and WH portions (domains as red boxes in Fig. 4a).

Figure 6a–c shows the decadal trend of global Zonal Mean Meridional Overturning Stream Function (ZMMSF) in response to FF, EastFF, and WestFF during 1980–2020. The ZMMSF in response to FF features a counter-clockwise Hadley Cell anomaly (shown in blue) over the tropics, which indicates a northward shift of the Hadley Cell into NH. The northward shift of Hadley Cell also occurs in response to WestFF, but not to EastFF, indicating that the shift of Hadley Cell is mainly due to

the WestFF. The global mean ZMMSF shifts in our results are consistent with previous studies (Xu and Xie, 2015; Allen and Ajoku, 2016; Amaya et al., 2018; Shen and Ming, 2018) focusing on the inter-hemispheric forcing gradient. That is, the tropical circulation always tends to move towards a warmer hemisphere with stronger radiative forcing.

To further diagnose why EastFF and WestFF induce distinct changes of the Hadley Cell, we analyze the changes of zonal,

column integrated meridional energy transport in response to aerosol forcings, which is shown in Fig. 7b–d. The atmospheric energy transport (AET) is calculated based on the:

$$\frac{\partial}{\partial \varphi} F_a = R_{TOA} - Q, \tag{2}$$

Where $\varphi$ is latitude, $F_a$ is the meridional energy flux, $R_{TOA}$ is the net radiative flux at the top-of-atmosphere (downward positive) and Q is the net downward energy flux at the surface. Q includes shortwave radiation, longwave radiation, sensible

heat flux, and latent heat flux. AET is then obtained by integrating the energy flux from south to north:

$$AET(\Phi) = 2\pi a^2 \int_{-\pi/2}^{\Phi} cos\, \Phi' \left(R_{TOA} - Q\right) d\Phi' \tag{3}$$

Where a is the Earth radius. Similarly, the oceanic energy transport (OET) is calculated based on the surface radiative flux:

$$\frac{\partial}{\partial \varphi} F_o = Q \tag{4}$$

The positive radiative forcing anomaly in NH extratropics from WestFF induces a negative AET at the equator (Fig. 7d),

which leads to the northward shifts of Hadley Cell and ITCZ to balance the interhemispheric difference in radiative forcing. Previous studies demonstrated that cooling NH leads to southward shift of ITCZ (Broccoli et al., 2006, Kang et al., 2021) consistent with the findings here. On the other hand, the EastFF introduces strong negative radiative forcing in the tropics and weak positive forcing anomaly in NH extratropics, but the AET shows small trends at all latitudes compared to that due to WestFF (Fig. 7c). Therefore, the Hadley Cell does not shift significantly in response to EastFF. The AET changes in

response to FF resembles that in response to WestFF, indicating the dominant role of WestFF in shifting the Hadley Cell.

Figure 6d–f shows the global-mean zonal wind (U) trends. The FF and EastFF forcing induce slowing U on the poleward flank of the NH jet core while strengthening U on the equatorward flank (especially in the EastFF case), indicating equatorward shifts of the NH jet stream. However, in the WestFF, U decreases on both flanks of the jet core (slighter greater on the equatorward flank) and the position of the jet core has no significant shift. This is not consistent with the shift of Hadley Cell. As a result, the shift of Hadley Cell and NH jet stream in FF case are in opposite directions, which appears to disagree with previous studies (Xu and Xie, 2015). However, based on the regional FF simulations (Fig 6b–c & e–f), we show that the jet stream and Hadley Cell in FF are controlled by different regional forcings during this period (EastFF drives jet stream shift vs. WestFF drives Hadley Cell shift), which agrees with the argument provided by Xu and Xie (2015). The competitions between EastFF and WestFF in shaping the Hadley Cell and NH mid-latitude Jet stream further indicate the importance of the meridional location of the aerosol forcings in addition to the zonal difference. Previous studies (Seo et al., 2014; Kang et al., 2021) also suggest the importance of latitudinal position of the radiative forcing to the movement of tropical circulations, which is consistent with our findings here.

The latitudinal profiles under the contour plots in Fig. 6 d–f indicate the corresponding zonal mean FSNTOA *gradient* trend (black curves; in the unit of W/m$^2$/decade/10ºLat), which seems to provide a good rule-of-thumb guidance for the expected shift of NH jet stream. The FSNTOA gradient in response to FF and EastFF show an increase of FSNTOA gradient trend over the mid-latitudes (black dashed lines), which is consistent with the equatorward shift of the NH jet stream. On the contrary, the FSNTOA gradient in WestFF shows a slight negative gradient, while the NH jet stream shows no significant shift. Note that the FSNTOA gradient is only a quick rule-of-thumb guidance of the NH jet shift, and one cannot explain the jet stream shift only based on it. More precise mechanisms of the jet stream shift driven by FF forcings will be discussed below.

Figure 8a–c shows the zonal-mean geopotential wind in the zonal direction (Ug) in EH (red dashed box in Fig. 4), which is derived from geopotential height (Z) following the geostrophic wind equation. The derived Ug patterns always resemble the simulated U pattern in EH (Fig. 8d–f), WH, and Global (not shown), revealing the strong correlation between tropospheric circulation changes and the tropospheric temperature changes (and thus the geopotential height changes). Instead of the gradient of radiative variables as in Fig. 6, here we show the latitudinal profiles of the trend of SAT gradient in response to each force. It has been previously demonstrated that the tropospheric responses to sulfate aerosol are anchored to the SST gradient (Xu and Xie, 2015).

The negative radiative forcing from EastFF is located at low latitudes, which leads to a negative SAT gradient southward of 35 ºN. The EastFF-induced SAT shows negative gradient on the equator flank of the NH jet core (southward of 35 ºN) and positive gradient on the polar flank (max at about 45 ºN), indicating a great strengthening of zonal wind on the equatorward flank and weakening on the poleward flank. This is producing a net effect of an equatorward shift of NH jet. On the other

hand, the WestFF-induced radiative forcing is located at mid-to-high latitudes, which leads to largely balanced radiation flux on both flanks of the jet core. Therefore, the WestFF-induced SAT does not show a significant gradient, and thus the jet stream shifts are much weaker compared to the FF and EastFF cases.

To summarize, Fig. 8 shows that the local trend of SAT gradient well explains the weakening or strengthening of the NH jet stream following the geostrophic wind equation, while the latitudinal slope of the SAT gradient (dashed line as the linear fit in Fig. 8) indicates the shift of jet stream. A consistent governing principle emerging from Fig. 8 is that: NH jet stream always tends to shift towards the more negative portion of SAT gradient. This is consistent with Fig. 6 – NH jet stream shifts towards the more negative portion of forcing gradient. Based on Fig. 7, the AET fails to explain the shift of NH jet stream,

indicating that the jet stream may be more controlled by the slow response of the aerosol forcing due to surface temperature change rather than the fast response.

As shown in Fig. 5, the EH still experiences a warming tendency in response to FF, the same as WH, despite an increasing aerosol emission locally at low-to-mid latitudes. So, to further reveal the contrast between EH and WH in response to FF as

well as regional FF forcings, we re-assess the identified relationship between radiative forcing gradient, temperature gradient, and the tropical circulation changes in Fig. 6–7, by extracting the regional signals from the global mean states.

Figure 8 (1st and 2nd rows) shows the regional ZMMSF and U changes relative to the global-mean state. The EH-Globe (EH minus global mean) and WH-Globe show opposite ZMMSF trends at low latitudes with opposite slopes of FSNTOA

gradients in both the FF case and the regional FF cases, indicating the importance of cross-equatorial AET to govern the tropospheric circulation adjustment. The increasing FSNTOA gradient from south to north in NH leads to a clockwise ZMMSF trend and a poleward shift of Hadley Cell, which is consistent with the result shown in Xu and Xie (2015) and references within.

The relationship between the gradient of the SAT trend and the shift of the NH jet stream is re-assessed in the middle row of Fig. 9, in terms of regional anomalies over EH and WH relative to the global average. The results also support the simple relationship we identified: the NH jet stream shifts to the flank with a more negative SAT gradient because the magnitude of U trend at certain latitudes is determined by the local gradient of air temperature trend, with a more negative gradient strengthening the westerly wind there. One Counterexample here is the WH-Globe U in response to WestFF, where the jet

stream does not shift even with a slight negative SAT gradient. The air temperature trend pattern (bottom row of Fig. 9) reveals that the negative gradient of air temperature locates at 30 ºN at all pressure levels, which is the latitude of the jet core. As a result, the jet core does not shift much as desired. In the low-to-mid latitudes of EH (5 ºN to 35 ºN), the gradient of the FSNTOA trend in response to FF is negative compared to the global-mean, which is largely consistent with the increasing aerosol emission in EH (3rd row of Fig. 2).

### 3.4 Surface temperature responses over the NH with a focus on North Pacific

Having demonstrated the tropical circulation changes and NH jet stream changes in Sect. 3.3, now we look at the SAT response to the regional aerosol forcings. Many previous studies have examined the relationship and mechanisms about Atlantic changes (Booth et al., 2012; Bellomo et al., 2018; Hua et al., 2019; Watanabe and Tatebe. 2019). In contrast, the aerosol effects on the Pacific Ocean are comparatively less studied in the previous work (Allen et al., 2014; Dong et al., 2014; Hua et al., 2018), and the potential effects of aerosol redistribution need further discussion. Since this study focuses on the comparison and competition of East and West aerosol forcings, we are specifically interested in how the increasing Asia aerosol forcing affects the North Pacific and how that might be compensated by declining aerosol forcing from North America.

The 1st row of Fig. 10 shows the 40-year linear trends of Surface Air Temperature (SAT) over the ocean in response to FF, EastFF, and WestFF during 1980–2020, which bears a close similarity to SST (not shown). Overall, FF forcing induces significant warming over the North Pacific northward of 40 ºN, which is even stronger than the North Atlantic warming but with weaker trend in TOA net radiative flux ($R_{toa}$; second row of Fig10). EastFF induces significant cooling over the western part of the North Pacific at low-to-mid latitudes, which is consistent with previous studies (Dong et al., 2014; Takahashi and Watanabe, 2016; Smith et al., 2016). In contrast, WestFF, with positive forcing anomaly at mid-to-high latitudes (30 ºN–60 ºN; blue oval in Fig. 10), induces large-scale warming locally at North Atlantic and even stronger warming over the entire North Pacific. Thus, the WestFF-induced warming over the North Pacific largely offsets the EastFF-induced cooling in the FF case.

The North Atlantic warming, as many other studies (e.g., Acosta Navarro et al., 2017; Qin et al. 2020) pointed out, can be attributed to the reduction of aerosol emission over North America and Europe since the 1980s, which is clearly seen in TOA net energy flux (blue circle in the 2nd row of Fig. 10). A North Atlantic warming hole is also significant in the FF response (Dagan et al., 2020; Fiedler and Putrasahan 2021). Notably, the simulated warming hole is less significant in response to WestFF forcing alone. In response to the EastFF forcing, the high latitudes of the North Atlantic show a warming trend despite insignificant local changes of TOA net energy flux (2nd row of Fig. 10).

Over the tropical Pacific region, EastFF induces an El Niño-like SAT pattern with symmetric warming trends (does not pass the 95% significance test, though). The EastFF-induced El Niño-like pattern contradicts some previous studies arguing that Asian aerosols lead to a La Niña-like pattern (Kaufmann et al., 2011; Smith et al., 2016; Kang et al., 2021). On the other hand, WestFF induces an asymmetric SAT pattern over the tropical Pacific, with warming in the north and cooling in the

south. The distinct tropical Pacific SAT responses due to EastFF and WestFF may also contribute to the Pacific decadal to multi-decadal variability (PDV) in amplitude and spatial pattern. The question about whether and how regional aerosol forcings affect PDV needs further investigation.


Let's compare the three sets of responses. The weak cooling over the Pacific warm pool region in response to FF can be explained by the offsetting effects between EastFF and WestFF, where the WestFF-induced warming weakens the strong cooling due to EastFF. Similarly, at the North Pacific region southward of 40 ºN, the extended aerosol cooling effect from East Asia is largely offset by the warming effect due to WestFF in the total FF response. A notable finding is that, at least

based on the simulation here, the North Pacific warming northward of 40 ºN is dominated by the positive forcing anomaly from WH mid-to-high latitudes, overwhelming the cooling from EH low-to-mid latitudes. The latitudinal difference between EH and WH forcing distribution plays an important role here. Indeed, Fig. 4 shows that the EastFF-induced CDNC changes are concentrated over low-to-mid latitudes (close to the emission sources and western subtropical Pacific). In contrast, the WestFF-induced CDNC changes expand to a larger domain over North Pacific and North Atlantic. The simulation here

indicates that mid-to-high latitude SAT is more sensitive to extratropical forcing than forcings originating from a lower latitude. This finding is consistent with previous findings that emission at higher latitudes generates stronger temperature responses (Shindell and Faluvagi, 2009; Persad and Caldeira, 2018). We also examine the BB case (not shown), which has a strong negative forcing over northeastern Asia over 50 ºN, and we find that BB-induced cooling occurs over the entire North Pacific similar to WestFF-induced response. Therefore, we highlight that the latitudinal distribution of aerosol forcing is

essential to the North Pacific climate responses.

The competition of EastFF and WestFF over the North Pacific deserves some more discussion. There is an apparent paradox in the FF case: compared to the North Atlantic, the high-latitude region of North Pacific shows stronger warming, though with much weaker $R_{toa}$ trend. This suggests that the $R_{toa}$ cannot fully explain the North Pacific warming. So how does WH

forcing lead to North Pacific warming (and also in the tropospheric circulations of EH mid-to-high latitudes in Fig. 9)? We now try to discuss the mechanisms of North Pacific SAT adjustment based on the meridional energy transport.

The trends of the zonal mean atmospheric meridional heat transport (ZMMHT) are shown in the third row of Fig. 10, where WH shows positive trends northward of 60 ºN in all three cases (note that the climatology of ZMMHT over both North

Pacific and North Atlantic are still positive in poleward direction, as shown in line contour). This suggests that the meridional energy transfer from the Atlantic Ocean to the Arctic is enhanced, slowing down the North Atlantic warming rate due to local positive radiative forcing anomaly. Conversely, North Pacific ZMMHT (poleward in climatology) is weakened and thus exports less heat from North Pacific mid-latitudes to the polar region (blue color in the EH panels of Fig. 10).

In addition to the atmospheric ZMMHT, we also show the zonal, column integrated AET response in EH versus WH (bottom row of Fig. 10). The AET results show a strong northward energy transport trend in the WH, while the equatorward energy transport trend in the EH, both of which resemble the ZMMHT trends. Moreover, the strong WestFF positive radiative forcing anomaly induces the strong poleward energy transport trend in the WH and the AET in response to EastFF is much weaker.


Previous studies have also shown that the surface temperature response to aerosol forcing is also strongly modulated by the oceanic energy transport (OET; Cai et al., 2006; Delworth and Dixon, 2006; Dagan et al., 2020; Menary et al., 2020; Fiedler and Putrasahan 2021; Hassan et al., 2021). Here we show that OET (blue lines in Figure 10 bottom row) can have different trends from AET or ZMMHT. In response to the WestFF (positive) forcing, OET increases in WH high latitude and

decreases in EH, indicating that poleward heat transport via the ocean is strengthened over the North Atlantic, but is weakened over the North Pacific. The stronger poleward OET in WH, in addition to poleward AET in a larger magnitude, explains why the North Pacific shows a stronger warming trend without local forcing. The EastFF also induces increasing OET in WH and decreasing OET in EH, but with small magnitudes compared with WestFF-induced OET responses, which is similar to AET responses. This further suggests that the WestFF forcing at mid-to-high latitude is the dominant driver of

atmospheric and oceanic poleward energy transport in NH.

To summarize, the simulated North Pacific warming northward of 50 ºN is dominated by the WestFF positive radiative forcing anomaly at mid-to-high latitudes. The EastFF forcing at low-to-mid latitudes, which are extensively discussed by previous studies, only induces cooling trend at mid-to-low latitudes (southward of 50 ºN) near the emission domain, which is

similar to the positive El Niño-like pattern response, but the signal is overwhelmed by the warming tendency induced by WestFF, via both radiative forcing response and meridional heat transport from pole region. Another possible explanation for the stronger North Atlantic warming is that the mid-latitude region of EH is controlled by the WH emission reduction effects at mid-to-high latitudes via the zonal energy transport between the Atlantic and Pacific (McGregor et al., 2014), but it is not tested yet in this study.


## 4 Summary

The main findings of this paper are:

(1)      The significant zonal and meridional contrasts in aerosol emission redistribution lead to opposite local radiative

forcing that has competing effects on temperature and circulation changes.

(2)        In terms of hemispheric surface temperature response: the overall FF emission decreases since the 1980s, mostly driven by the WH, induces a significant warming trend over NH. Interestingly, although the FF emission over EH continuously increases, the SAT in EH still shows a warming trend as large as WH, because of the heavy influence from WH aerosol reduction.

(3)        In terms of tropospheric circulation responses: in response to FF, NH shows an overall positive gradient of temperature trend (cooling low latitude, warming high latitude), inducing a poleward shift of Hadley Cell and an equatorward shift of NH jet stream (Fig. 6). Previous studies show that the shift of mid-latitude jet stream is associated with the SST meridional gradient, and a cooling NH drives an equatorward shift of both Hadley Cell and NH jet stream (Xu and Xie, 2015; Xu et al., 2016), which appears to be inconsistent with our results. The main difference is that previous results are largely based on a sharp interhemispheric forcing gradient with a time scale of the entire 20th century (Wang et al., 2020a), or focusing on the mid-century era when aerosol emission is on the rise globally. In this study, focusing on 1980–2020 trend, the regional FF aerosol forcing competitions *within* the NH are more complex due to both zonal and meridional differences. The competing effects of EH and WH aerosol forcings are examined with two sets of large-ensemble simulations (Fix_EastFF1920 and Fix_WestFF1920) that we conducted.

In the tropics, the WestFF forcing at mid-to-high latitudes dominates the northward shift of the Hadley Cell. WestFF induces positive forcing anomaly in the NH extratropics and drives the northward shift of the Hadley Cell to balance the interhemispheric difference in radiative forcing. Interestingly, in the extratropics, the EastFF forcing at low-to-mid latitudes dominates the equatorward shift of the NH jet stream. The gradient of FSNTOA provides a rule-of-thumb guidance for the expected shift of the NH jet stream (Fig. 9), where the NH jet stream tends to shift to the negative gradient portion of FSNTOA. The competitions between EastFF and WestFF aerosol forcings in shaping the Hadley Cell and mid-latitude jet stream highlight the critical role of the meridional location of the aerosol forcings within the NH.

(4)        In terms of North Pacific temperature response, the FF forcing during 1980-2020, unlike suggested by previous studies, induces North Pacific and pan-Pacific warming due to a competition between EastFF and WestFF, with the latter dominating the former (Fig. 10). The dominance is due to the latitudinal distribution of aerosol forcing within the NH. The negative forcing of EastFF, which occurs at the EH tropical and subtropical region (0–40 ºN), is largely confined to the emission domain. In contrast, negative FF forcing over WH mid-to-high latitudes (northward of 30 ºN), not only introduces local warming but also imposes a heavy influence over North Pacific warming. Diagnostic data shows that the remote contribution to North Pacific warming from WestFF is due to the combination of both radiative forcing responses and meridional energy transfer anomaly from the North Atlantic to North Pacific via the Arctic pathway.

The importance of the inter-hemispheric asymmetry of external forcing has been extensively discussed in previous studies (e.g., Xu and Xie, 2015; Chung and Soden, 2017; Wang et al., 2020a), in which the NH is usually considered as a whole. Here we further emphasize that the latitudinal difference of external forcing within the NH is important in determining the tropospheric and surface responses. More specifically, the EastFF induces a PDO-like pattern (negative phase) with a cooling over the northwestern Pacific and warming over the tropical Pacific. However, when combined with the aerosol

reduction over the WH, the EastFF-driven northwestern Pacific cooling is completely offset and instead exhibits a warming trend.

We provide the following discussions related to our conclusion and suggest some future research directions.

(1)    The issue of nonlinearity. The CESM1 single forcing large ensemble simulations applied in this study treats the FF and BB forcing separately, which enables many of our discussions above. However, one problem is the additivity of the two anthropogenic aerosol sources. Deser et al. (2020) sum up the FF, BB, and GHGs responses to reconstruct the all forcing (ALL) response and find some inconsistencies due to the nonlinear interactions between aerosol- and GHG-induced responses. So, it is also possible that adding FF and BB may cause some nonlinear problems. Similarly, adding EastFF and

WestFF responses may also cause such problems. This study is not affected by such additivity problems because we treat each set of simulations separately, but future studies that aim to utilize the combined AA-induced changes, should be cautious and a rigorous test on additivity would be necessary.

(2)    The issue of RCP8.5 scenarios applied in CESM1. As is described in Sect. 2.1, the RCP8.5 scenarios are applied in

all five sets of large ensemble simulations used in this study for 2006-2020. Studies have pointed out that the RCP dataset is different from the new SSP scenarios which contain the historical emission data up to 2015. For example, the sulfate emissions over India and China over the 2006-2020 period in the RCP8.5 scenarios are lower than observations and SSP scenarios, which could lead to an underestimate of the sulfate cooling effects locally and globally (Lin et al., 2018). Future experiments utilizing the newer GCMs and SSP scenarios would be helpful to repeat and test the current results.


(3)    The single model large ensemble method. The single model large ensemble simulations applied in this study effectively separate the external forcing induced response from the model internal variability (Kay et al., 2015; Deser, et al., 2020). However, one limitation here is that only one GCM is utilized, so the potential systematic errors from a single model cannot be tested. Some new single-forcing large ensemble simulations based on models other than CESM1 (e.g., CanESM2

(Oudar et al., 2018); CESM2) are also coming out recently as part of the "Single Model Initial-condition Large Ensemble" (SMILE) efforts, which brings the possibility of testing "multi-model large ensemble" in future works and improving our understanding of aerosol induced climate change.

(4)    The implication of the North Pacific response on the PDO mode and global warming rate. The increasing FF
emission over EH low latitudes (without WH) induce a PDO-like SAT response over the North Pacific and tropical Pacific,
suggesting the potential impact of anthropogenic forcing onto the internal variability. More future research is needed on this
topic based on the current large ensemble simulations and the upcoming CESM2 simulations.

(5)    The difference of EastFF and WestFF in the north-south direction too. It is clear that the redistribution of
anthropogenic aerosols since the 1980s is not a pure zonal shift, but also shifts in the meridional direction. The aerosol
loading shows a net decrease, especially in the mid-latitude/subpolar region. Future studies may consider both zonal and
meridional shifts in SO4 loading.

**Data Availability**

The CESM1 Large ensemble data sets are publicly available from https://www.cesm.ucar.edu/projects/community-projects/LENS/data-sets.html (Kay et al., 2015). The CESM1 "Single Forcing" Large Ensemble data sets are publicly available from https://www.cesm.ucar.edu/working_groups/CVC/simulations/cesm1-single_forcing_le.html (Deser et al., 2020). The outputs for two sets of "regional" single forcing large ensemble (Fix_EastFF1920 and Fix_WestFF1920) are available on the NCAR Campaign Storage file system at /glade/campaign/univ/utam0012/CESM11-LEN-Reg/, and can also

be accessed via NCAR Data Sharing Service Endpoint on Globus upon request to authors.

**Author contributions**

CD and YX developed the idea for this study and designed the model experiments. CD performed the model simulations and data analysis, with input and feedback from YX and SX. CD and YX prepared the manuscript with contributions from all authors.

**Competing interests**

The authors declare that they have no conflict of interest.

**Acknowledgments**

We acknowledge National Center for Atmospheric Research (NCAR) for high-performance computing support from Cheyenne (doi:10.5065/D6RX99HX) and data storage resources provided by Computational and Information Systems

Laboratory (CISL). We thank the CESM-LE project for providing access to model outputs. Diao and Xu acknowledge funding support from the US National Science Foundation's Climate and Large-scale Dynamics Program (AGS-1841308).

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

**Figures**

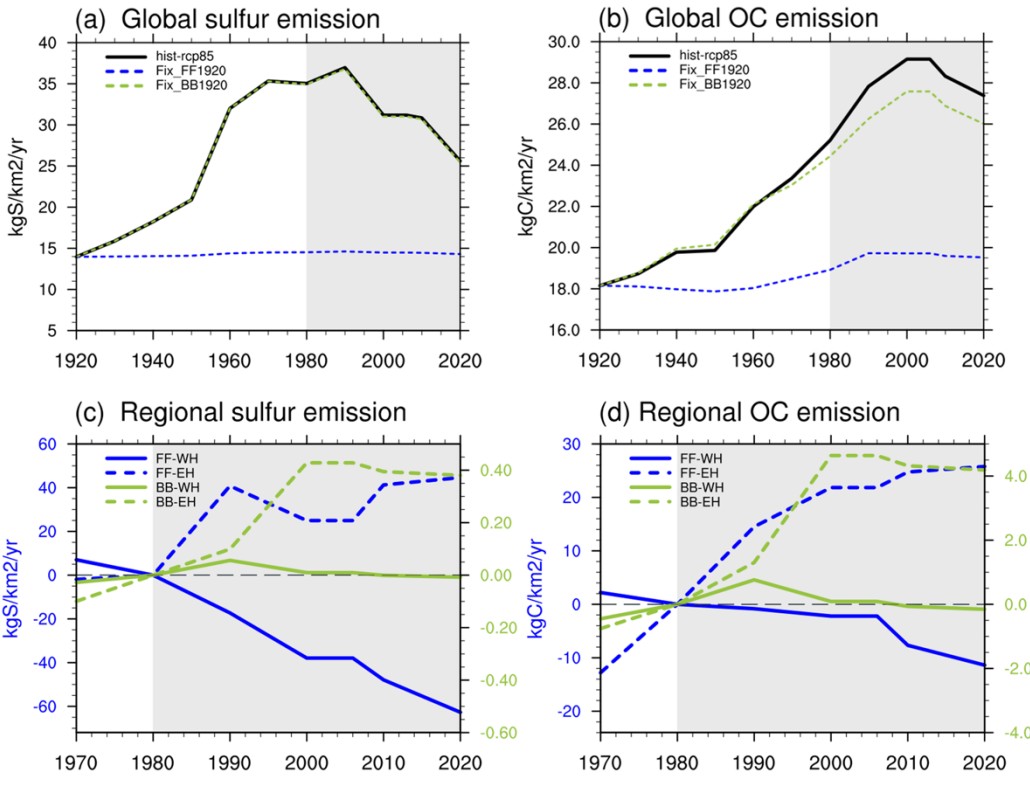


**Figure 1:**

(a) & (b): The global average anthropogenic emission of Sulfur and OC from 1920 to 2020 in the three sets of simulations used in this study. The "FF" represents the emission-related to Fossil Fuels. The "BB" represents Biomass Burning. Shading areas of 1980-2020 are the focused period.

(c) & (d): Similar to (a) & (b) but as the difference between ALL and corresponding fixed aerosol experiments to show the regional FF or BB emission. FF (blue lines) uses the left-hand side Y-axis and BB (green lines) uses the right-hand side Y-axis. Solid lines are for the "Western box" (West-box; 0-80ºN, 120ºW-40ºE), and dashed lines are for the "Eastern box" (East-box; 0-80ºN, 60ºE-150ºE). Boxes are shown in Fig. 2. All numbers in (c) and (d) are relative to the 1980 level to illustrate the change from 1980 to 2020.

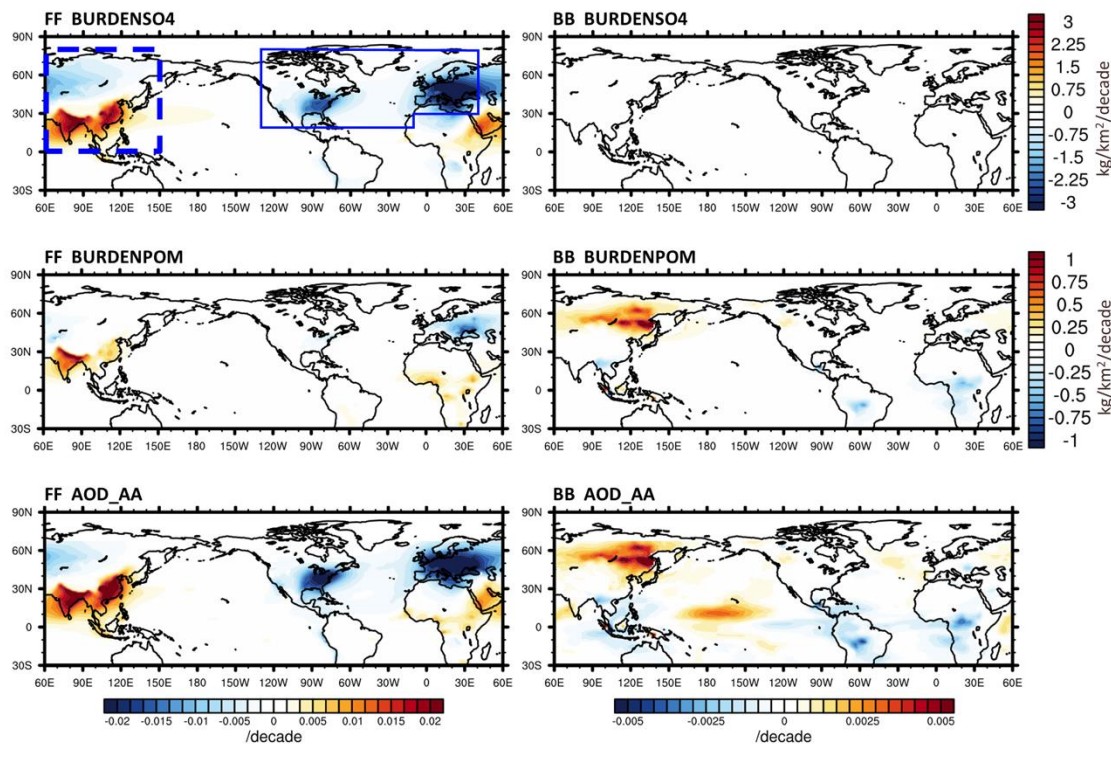


**Figure 2:**

**1st & 2nd row : 40-year trend of Sulfate and Primary OC (POM) column burden (kg/km2/decade) in response to FF (left) from 1980 to 2020, and BB (right) emission changes. BC and SOA are not shown but the patterns are very similar to SO4.**

**3rd row: 40-year trend of AOD-AA (aerosol optical depth due to anthropogenic aerosols). Note the color scale of FF is twice of BB.**

**The blue dashed box in the upper left panel indicates the "Eastern Hemisphere" box (EH-box, 0º–80ºN, 60º–150ºE) used for the regional fixed-aerosol simulations. The solid box indicates the "Western Hemisphere" box (WH-box, 20º–80ºN, 130º–10ºW, and 30º–80ºN, 10ºW–40ºE).**

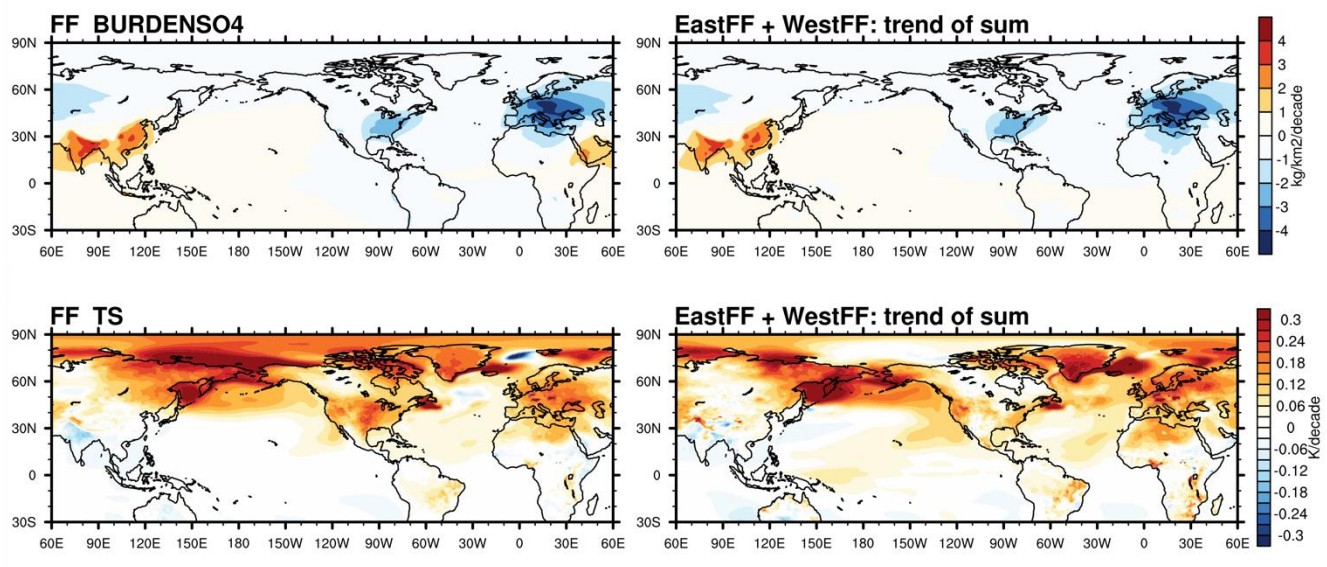

**Figure 3:**

Left column: the 40-year trend of (top) sulfate column burden (kg/km2/decade), and (bottom) surface air temperature in response to FF forcing from 1980 to 2020.

Right column: Same as the left column but for the summation of EastFF and WestFF, obtained by first adding EastFF and WestFF responses together and then calculating the 40-year trend.

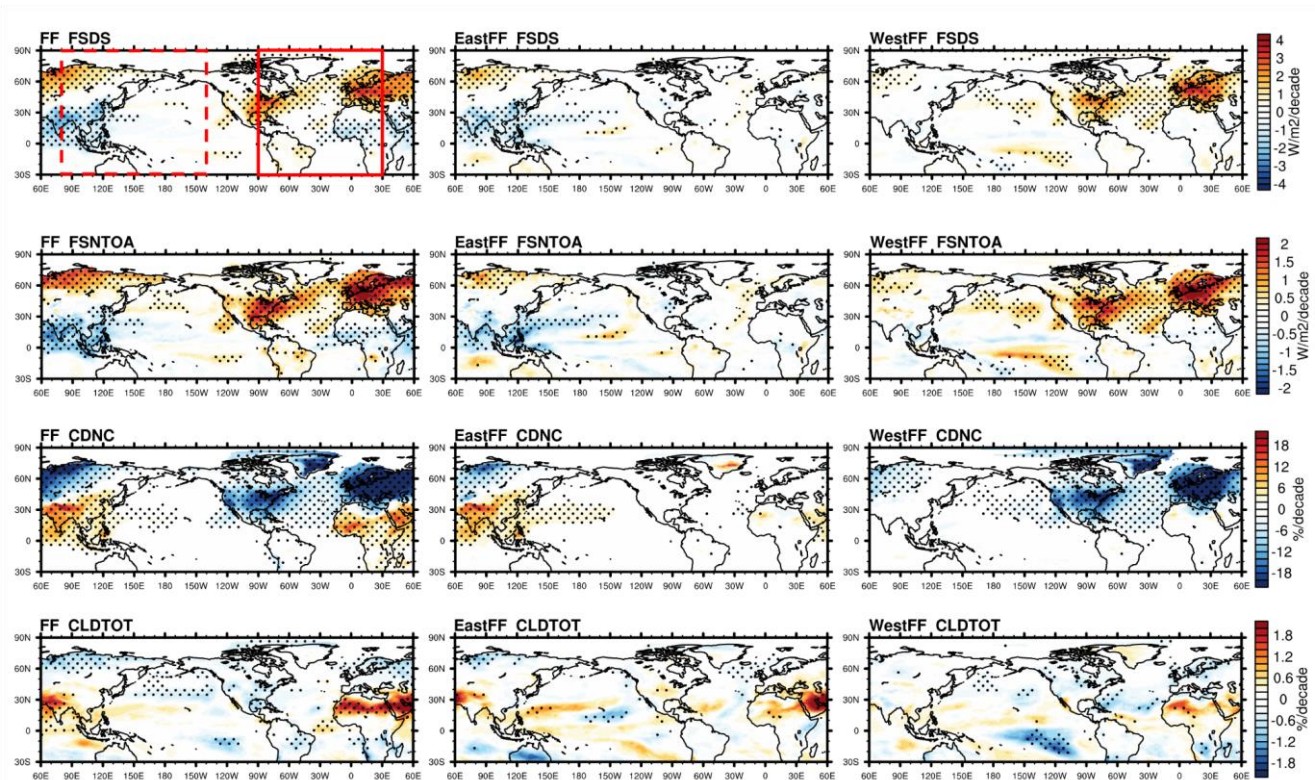


**Figure 4:**

(1st row) 40-year trend in surface downward solar flux (FSDS, W/m2/decade) in response to FF (left), EastFF (mid), and WestFF (right); (2nd row) Top-of-Atmosphere net solar flux (FSNTOA, W/m2/decade); (3rd row) vertically integrated droplet number concentration (CDNC, %/decade); and (4th row)Total cloud fraction (CLDTOT, %/decade). The CDNC and CLDTOT patterns
show the percentage trend relative to the 40-year climatology.

Dotted areas indicate the region where the trend passes the 95% significance test.The high latitude ocean regions with significant sea ice change (e.g., the Arctic and the Sea of Okhotsk) are masked to remove the surface albedo change effects. The dashed box in the left panel of 1st row indicates the "Eastern Hemisphere" (EH box, 80ºE-140ºW) used in our subsequent analysis, and the solid box indicates the "Western Hemisphere" (WH box, 90ºW-30ºE).


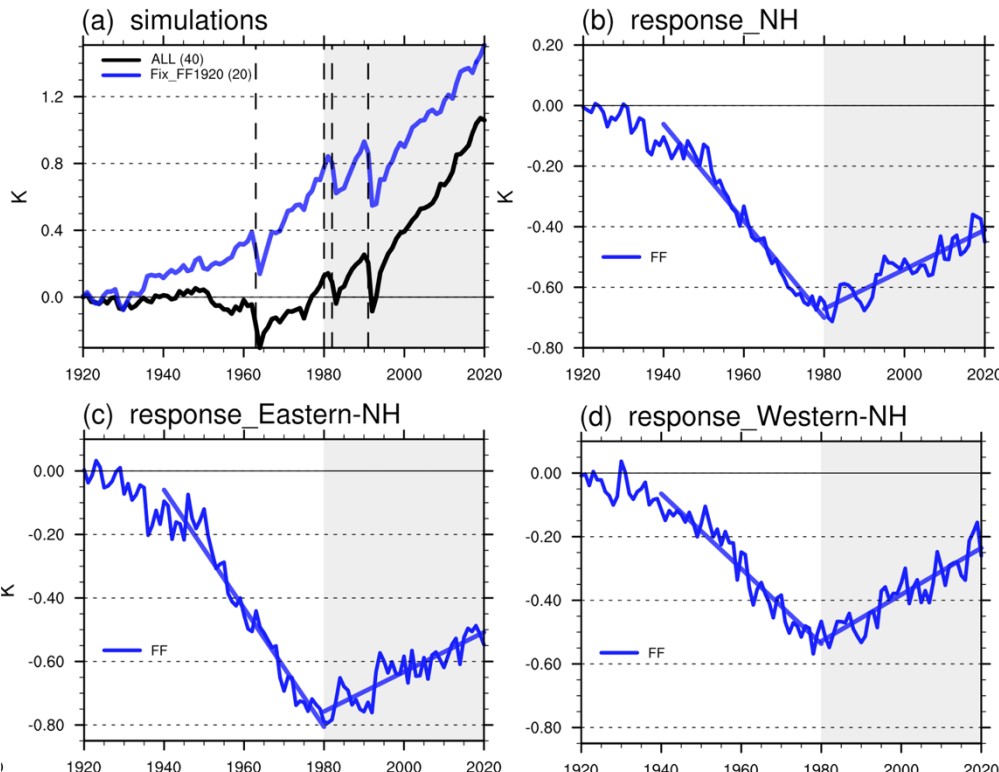

**Figure 5:**

**(a) The Northern Hemisphere (NH) surface air temperature (SAT) anomalies (relative to 1920) in two sets of simulations: ALL (all forcing: historical+RCP8.5) (black line; 40-member), and Fix_FF1920 (blue line; 20-member). The four dashed lines indicate the four major volcanic eruption events (greater than Category 5) during the 20th century. Shaded areas are the focused period of this study.**

**(b) The SAT responses to FF (blue). The thinker straight lines indicate the 40-year linear fits for 1940-1980 and 1980-2020 respectively. (c) & (d) are similar to (b) but for the EH and WH boxes (see boxes in Fig. 4).**


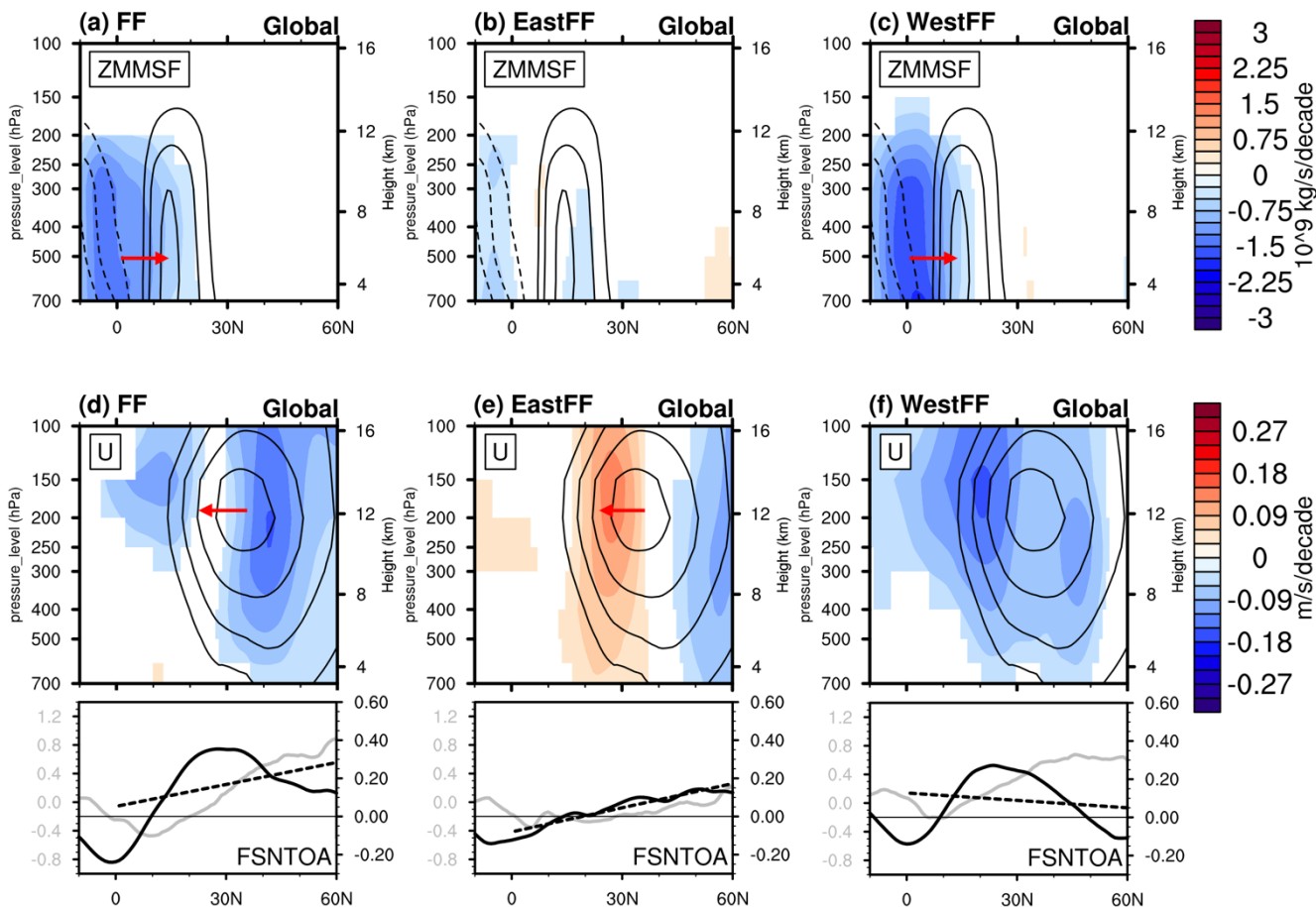

**Figure 6:**

(a–c) The 1980-2020 trend of zonal mean meridional stream function (ZMMSF, 10^9 kg/s/decade) in response to (a) FF, (b) EastFF, and (c) WestFF. The positive values (solid lines for climatology and red shading for long-term trend) indicate clockwise circulation. The positive values indicate westerly wind. The statistically insignificant trend is masked in white. The arrows indicate the latitudinal shift of Hadley Cell.

(d–f) Similar to (a–c) but for zonal mean zonal wind (U, m/s/decade). The latitudinal profiles below show the Top-of-Atmosphere net solar flux trend (FSNTOA as in Fig. 5c; W/m²/decade). The arrows indicate the latitudinal shift direction of the NH jetstream core.

The profiles below show the 40-year trend of TOA net solar flux (FSNTOA; grey curves; in units of K/decade), and the gradient of FSNTOA trend (black curves; in units of W/m2/decade/10°lat). The trend and gradient lines are smoothed using the moving average 925 method (with 30 degrees of latitudinal range sampling window). The dashed lines are the linear fit from 0° to 60°N.

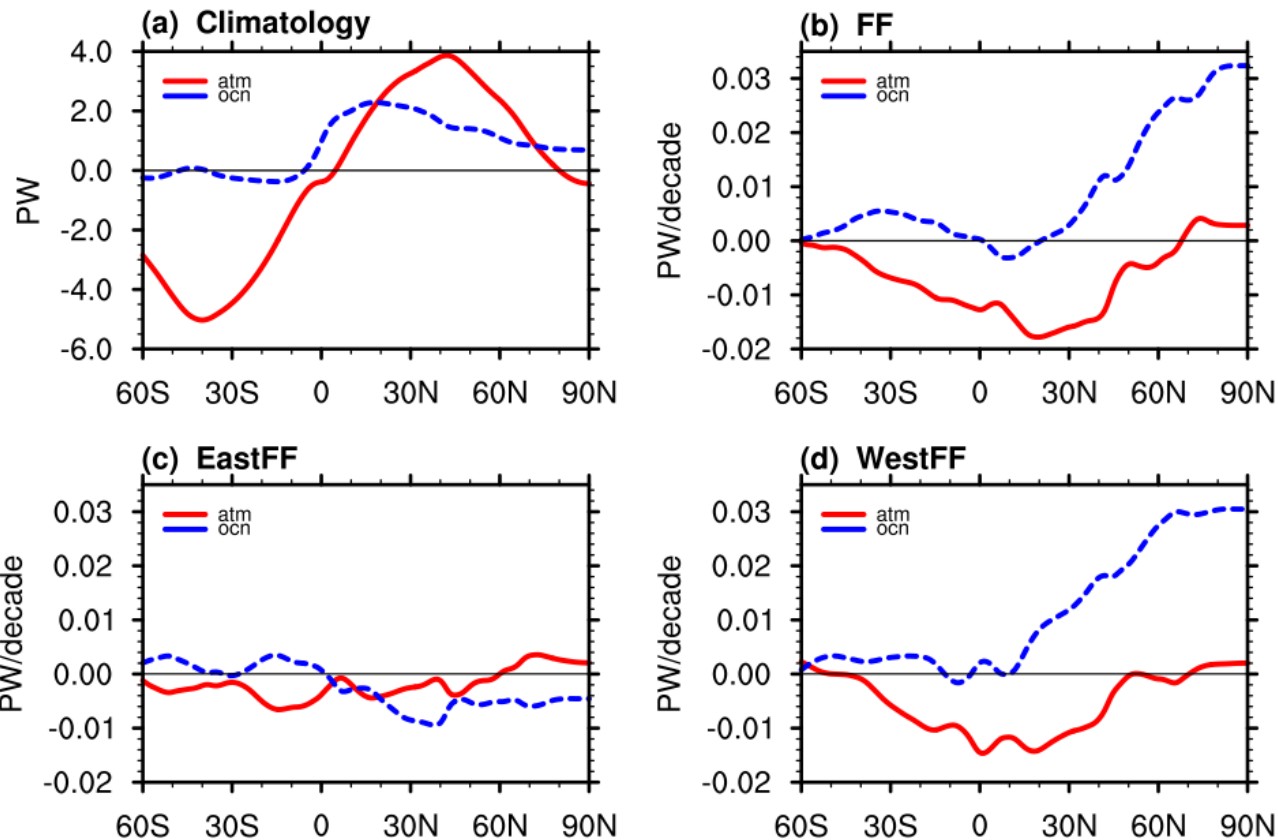

**Figure 7:**

(a) The 40-year climatology of northward energy transport (Pwatt) is calculated based on ALL experiments' ensemble average. (b–d) The decadal trend of northward energy transport in response to FF, EastFF, and WestFF (Pwatt/decade), which are obtained by subtracting the fixed single forcing experiments from the ALL experiment.

The dashed blue lines represent the oceanic energy transport; the solid red lines represent the atmospheric energy transport.


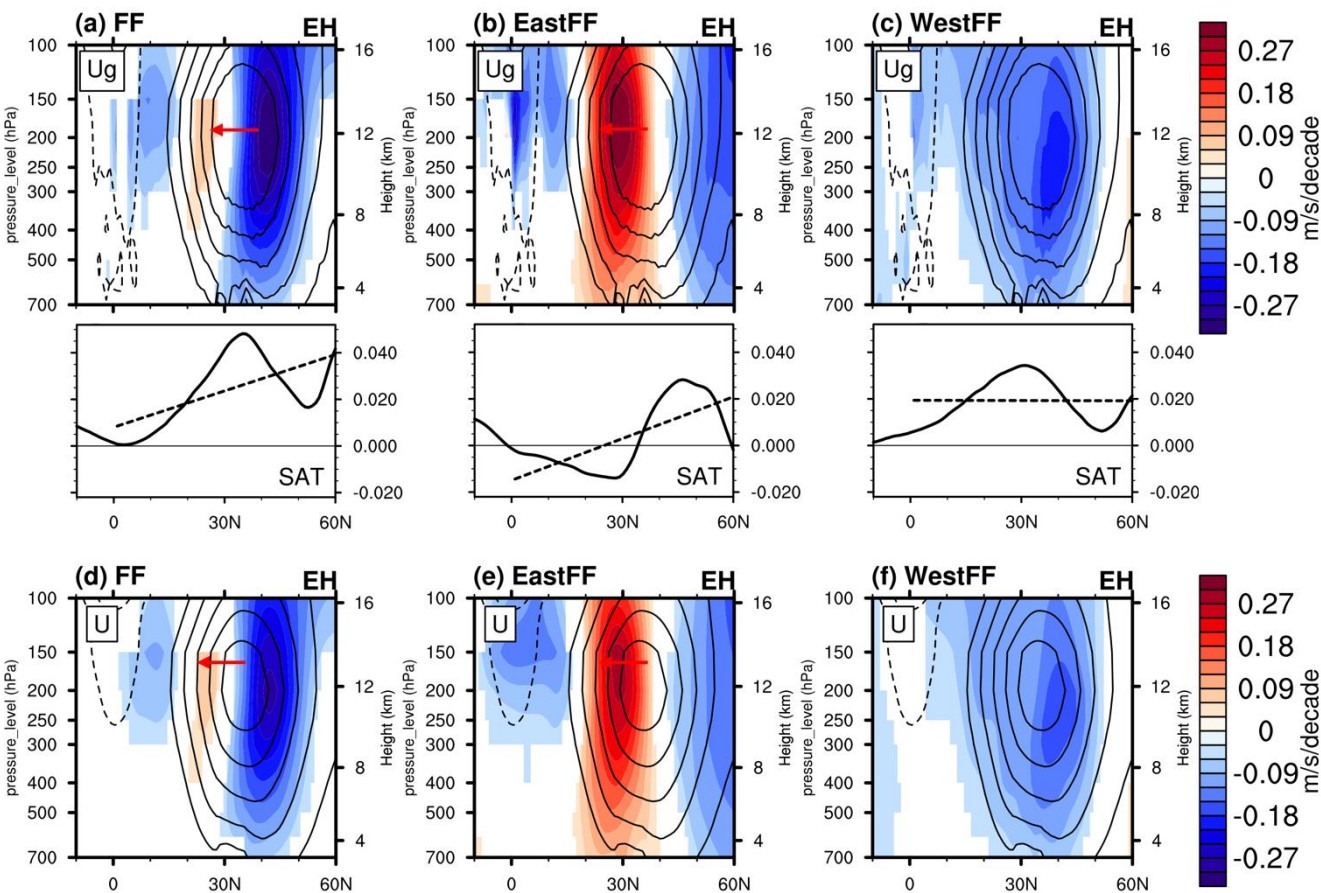

**Figure 8:**

(a–c) Similar to Fig. 6d–f but for geostrophic wind (Ug) trend (m/s/decade) derived from geopotential height (Z) during 1980-2020 in Eastern Hemisphere (EH). The climatology is shown in contour lines with an interval of 6. The positive values indicate westerly wind. The statistically insignificant trend is masked in white. The arrows indicate the latitudinal shift in each case.

The profiles below show the gradient of surface air temperature trend (black curves; in units of W/m2/decade/10ºlat). The gradient lines are smoothed using the moving average method (with 30 degrees of latitudinal range sampling window). The dashed lines are the linear fit from 0º to 60ºN. Positive values indicate a larger SAT trend at higher latitude, and vice versa.

(d–f) Similar to the bottom row of Fig. 6 but for the U trend in the Eastern Hemisphere (EH) in response to three types of forcings.

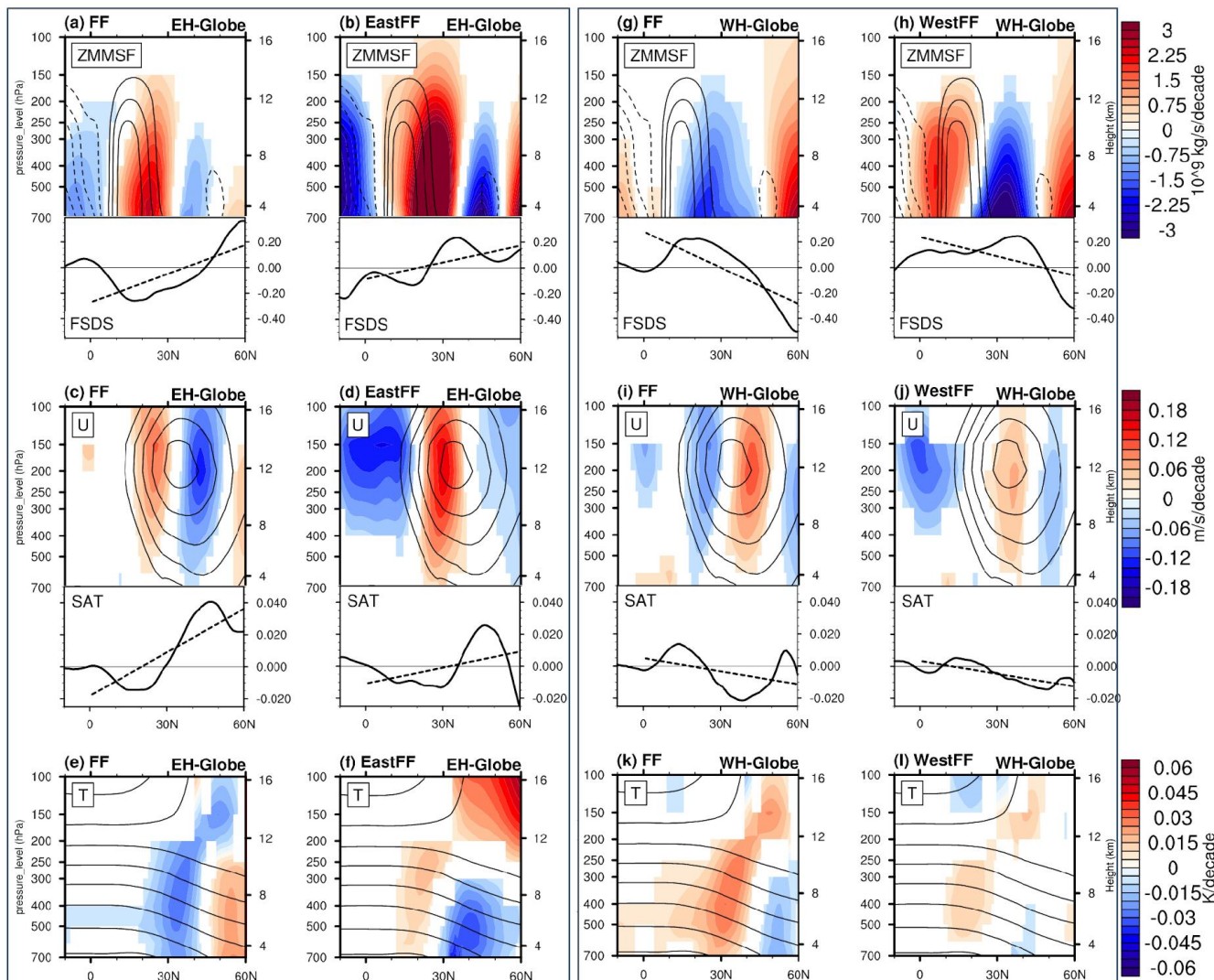

**Figure 9:**

**1st row: ZMMSF trend, Same as Fig. 6 (a–c) but showing the difference between global mean and EH mean (a–b), and the difference between global mean and WH mean (g–h). The latitudinal profiles below show the latitudinal gradient of the FSDS trend, which is also calculated as the difference between EH/WH and Globe.**

**2nd row: U trend, Same as Fig. 8 (d–f) but showing the difference between global mean and EH (WH). The latitudinal profiles below show the latitudinal gradient of SAT trend, which is also calculated as the difference between EH (WH) and Globe.**

**3rd row: Similar to the above two rows but showing tropospheric air temperature (T) trend.**

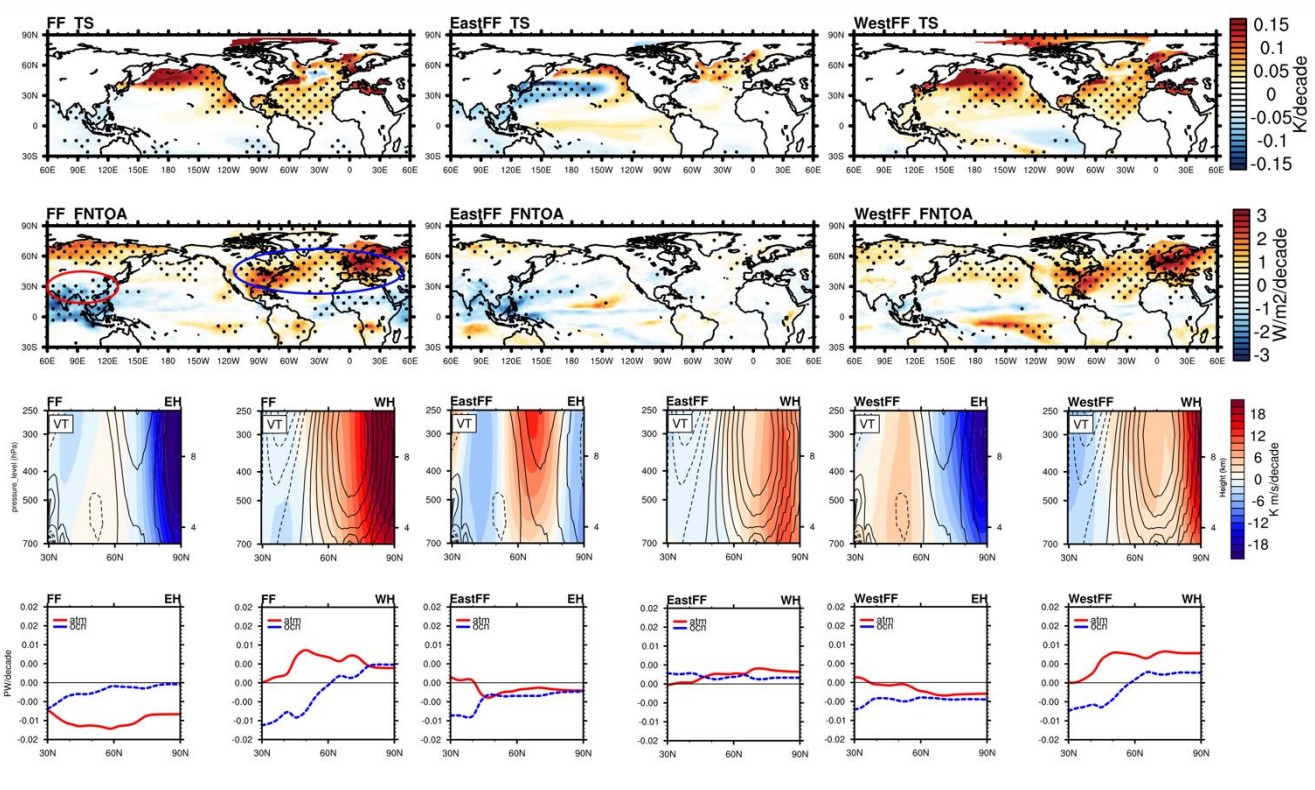


**Figure 10:**

**1st & 2nd rows: 40-year sea surface air temperature trend (SAT) and net energy flux at TOA (R$_{toa}$). Arctic regions with significant sea ice changes are masked.**

**The blue (red) circles in the 2nd row indicate the major regions with a decline (increase) of aerosol emission.**

**3rd row: 40-year trends of zonal mean meridional heat transport (ZMMHT) in EH and WH. The positive values (solid lines for climatology and red shading for long-term trend) indicate poleward transfer. The arrows indicate the energy transfer responses relative to the climatology.**

**4th row: Similar to Fig. 7 but showing the regional northward energy transport (Pwatt/decade; northward positive) in EH and WH. The dashed blue lines indicate oceanic northward energy transport (OET), while the solid red lines indicate atmospheric northward**
**energy transport (AET).**