# Peer review of "Anthropogenic Aerosol effects on Tropospheric Circulation and Sea Surface Temperature (1980-2020): Separating the role of Zonally Asymmetric Forcings"

_Atmospheric Chemistry and Physics, 2021_

## Referee Comment (RC2)

This study compared the radiative effects of anthropogenic aerosols between western hemisphere and eastern hemisphere in the recent decades and their impacts on the atmospheric circulation. It is an interesting topic, as the interhemispheric contrast of anthropogenic aerosols were often focused in previous studies but the eastern-western hemispheric contrast of anthropogenic aerosols gains much less attention despite this may be the dominant change of anthropogenic aerosols in the NH during the last four decades. The analysis is solid, but some descriptions are not very accurate and some mechanisms are still unclear. Moderate revisions are needed to address these concerns.

Detailed comments:
1. Page 2 L23-26: I don't think you can say the SAT gradient can be used as a **_predictive_** metric of NH jet changes, as they are just the same thing based on the thermal wind relationship. Both of them are forced by other factors.

2. Page 2 L28-29: First, this sentence seems problematic; Second, when you say "dominating role of WH forcing", which aspects do you point to? Based on your results, at least for the jet shift, it is more caused by EH forcing.

3. L38-40, IPCC AR6 shows it is 1.1C

4. L40-42: Dong and Mcphaden (2017) also shows a good example of how the GHGs and internal variability, such as IPO, shapes the global warming at the decadal time scale.
Lu Dong and Michael J McPhaden 2017 Environ. Res. Lett. 12 034011.

5. L53-55: Similar conclusions have been obtained in earlier studies, such as Salzmann (2016).
Salzmann, M. Global warming without global mean precipitation increase? Sci. Adv. 2, e1501572 (2016).

6. L44-61: Although there are many differences between GHGs and aerosols as you mentioned here, some other studies found some climate impacts caused by GHG and aerosols can be very similar. For example, Xie et al. (2013) showed the SST and ocean precipitation response patterns are very similar in both GHGs and aerosols. Recently, Song et al. (2021) also shows both GHG and aerosols modulate the seasonal delay of tropical rainfall in a similar way, i.e., by modulating the atmospheric column humidity. This similarity between the two forcings should also be mentioned.

Xie, SP., Lu, B. & Xiang, B. Similar spatial patterns of climate responses to aerosol and greenhouse gas changes. Nature Geosci 6, 828–832 (2013).
Song, F., Leung, L.R., Lu, J. et al. Emergence of seasonal delay of tropical rainfall during 1979–2019. Nat. Clim. Chang. 11, 605–612 (2021).

7. L59-61: As mentioned above, Song et al. (2021) found the recent decreases of aerosols, combined with the increased GHGs, contribute significantly to the seasonal delay of tropical rainfall. As the decreased aerosol and increased GHG will continue in the future, the seasonal delay of tropical rainfall is expected to amplify in the future.

8. L69: for the first reference: Who?

9. L78: tropics->tropical

10. L166-167: Fix_FF1920 have 20 members, but here Fix_EastFF1920 and Fix_WestFF1920 only contain 10 members, is there any sensitivity of the results to the member numbers? For example, if you also only use 10 members of Fix_FF1920, could you obtain the similar results? Another relevant question is that you should also show whether the trend of many variables you focused here in the Fix_FF1920 is roughly the sum of Fix_EastFF1920 and Fix_WestFF1920.

11. L201-202: We know the decreased trend of FF-related aerosols in the North America and Europe is due to the clean air acts and increased trend of FF-related aerosol in the India and China is due to the economic development, but what's the reason of the stronger increasing trend of BB-related POM over the northeastern Asia?

12. L217: removing "1"

13. L241: and FF-> FF and.

14. L238-240: Could you explain a little bit more about how the indirect effects of aerosols (i.e., cloud droplets number and cloud lifetimes are enhanced) could expand the affected regions? Do you mean the cloud formed in the emission region can be transported to other places?

15. L244: should be decrease of CLDTOT rather than increase in response to WestFF based on Fig. 3?

16. Figure 3. You mentioned that regions passing the 95% significance is dotted, but I didn't see any dots there. You may also need to do the significance test in Fig. 2 and many other figures.

17. Fig. 5: What did you do when you say "smoothed in 30 degrees of latitudinal range"

18. would you like to mention how the increased FSNTOA gradient drives the equatorward shift of the NH jet stream?

19. L371: below 35N? seems problematic. Suggest changing to southward of 35N

20. Fig. 6: Here, why do you only focus on the EH, rather than Global in previous figures? Could you explain it a little bit?

21. L394: sometimes using FSDS, but in other cases, you use FSNTOA. Please justify your choices.

22. L418: references are needed here.

23. L430: induce es->induces a

24. L468: other ver?

---

## Author Comment (AC1)

Review of:

Anthropogenic Aerosol effects on Tropospheric Circulation and Sea Surface Temperature (1980-2020): Separating the role of Zonally Asymmetric Forcings

By Diao et al.

This paper utilizes the CESM large ensemble simulations and its single forcing simulations component to study aerosol effect on surface temperatures and atmospheric circulation. In addition, two new sets of ensembles of simulations with the CESM are presented. These simulations are designed to capture the shift in recent decades in aerosol forcing from the western hemisphere mid-high latitudes (USA and Europe) to the eastern hemisphere low-mid latitudes (mostly India and Chania). The new simulations are well executed set of numerical experiments, which presents a valuable contribution to the climate community. I agree with the authors' statement that: "The zonally heterogeneous changes in AA emissions since the late 20th century, …, received less attention".

**Response:**

**Thanks for a precise summary of our study, which is mainly focused on the zonal heterogeneous forcing due to anthropogenic aerosols.**

**The new simulations will be released upon publication to the research community. See the updated data availability statement for the access method.**

However, I fill that this paper fails to present the current knowledge in the field.

**Response:**

**Thanks for the suggestion. We have added more references to the previous studies as suggested in the revised Introduction section. See detailed response below.**

In addition, the paper is mostly descriptive and the physical explanations are lacking (please see more details below).

**Response:**

**Thanks for the valuable comments. We have conducted further physical analyses as suggested, specifically for the linkage between the radiative imbalance and the Hadley circulation shifts. Please check our response to the major comments below.**

Finally, the paper could benefit from a carful re-writing and editing as some parts are not writen well and I found numerous typos.

**Response:**

**Thanks for the detailed suggestion in the specific comments. We carefully re-edited our manuscript and corrected many editorial issues.**

Having said that, I believe that after a major revision, this paper could make important contributions to understanding aerosols' climatic impacts.

**Response:**

**The current draft went through a major revision following valuable suggestions from the two reviewers.**

**General comments**

- As was mentioned above currently the paper is mostly descriptive and so little physical insight is gained. Specifically, the explanation of the circulation response is lacking. I have outlined below a few examples but there are defiantly more. Hence, I would like to encourage the authors to conduct deeper thinking and to try to bring the dynamic discussion to a level suitable for publication.

**Response:**

**Thanks for the valuable feedback. We have included more dynamical analysis to the manuscript in this round of revision:**

- **The cross-equatorial atmospheric energy transport due to radiative forcing occurring at mid-to-high latitudes leads to the Hadley Cell shifting northward to balance the interhemispheric energy imbalance (Fig 7).**

- **Both atmospheric and oceanic meridional energy transport over the North Atlantic contributes to the remote warming over the North Pacific (Fig 10).**

- **Analysis of the cloud droplet concentration to show the contribution of aerosol indirect forcings (Fig 4).**

- **The additivity and sensitivity test of the two sets of regional forcings experiments (EastFF and WestFF) (Fig. 3)**

  **Please check the detailed responses to the specific comments below.**

- The introduction is lacking many previous studies that examined the aerosol geographical distribution effect on circulation, forcing and temperature. Although I agree that this issue should get more attention, from the current manuscript it could sound almost like it is the first time anyone looked at it. This is not the case. A few examples of papers which come into my mind are Fiedler et al., 2017; Chemke and Dagan 2018; Persad and Caldeira 2018 and Fiedler and Putrasahan 2021.

   **Response:**

   **Thanks for the suggestions. We added more references to the Introduction section to summarize the previous works as suggested, some of them focusing on the zonal differences of aerosol forcing. The Introduction is also restructured for clarity. Apologies for the previous oversights.**

- In addition, the introduction focusses quite a bit on extremes. As this is not the focus of this paper, it is unclear to me why.

**Response:**

We mentioned a few previous studies related to the aerosol effect on extremes to highlight the broader implication of the inferred circulation changes here, although the paper itself here does not directly address extreme events.

We still like to keep these few (2-3) references in the Introduction but we have rearranged the Introduction for clarity, where the extremes-related discussion is now in a paragraph summarizing previous studies on regional aerosol forcing and climate responses.

**Specific comments**

L 13-14: this sentence sounds a bit awkward. Consider re-writing.

**Response:**

Thanks for the suggestion. We have modified the sentence as follow:

*"The overall cooling effects of AA, which mask a portion of global warming, have been the subject of many studies but large uncertainty remains."*

L 92. It defiantly also played a role in the north Atlantic. The motivation of focusing only on the north Pacific is not clear.

**Response:**

**Thanks for the suggestion. We agree that the AA forcings also affect the North Atlantic. We acknowledge that many previous studies have examined the relationship and mechanisms of Atlantic changes (Booth et al., 2012; Bellomo et al., 2018; Hua et al., 2019; Watanabe and Tatebe. 2019). In contrast, the aerosol effects on the Pacific ocean are comparatively less studied in the previous work. Also, since this study focuses on the comparison and competition of East and West aerosol forcings, we are specifically interested in how the increasing Asia aerosol forcing induces warming in the North Pacific and how that might be compensated by declining aerosol forcing from North America. Therefore, in section 3.4 we specifically focused on the North Pacific and we provided further motivation at the beginning of section 3.4, copied below for references:**

*"Having demonstrated the tropical circulation changes and NH jet stream changes in Sect. 3.3, now we look at the SAT response to the regional aerosol forcings. Many previous studies have examined the relationship and mechanisms about Atlantic changes (Booth et al., 2012; Bellomo et al., 2018; Hua et al., 2019; Watanabe and Tatebe. 2019). In contrast, the aerosol effects on the Pacific ocean are comparatively less studied in the previous work (Allen et al., 2014; Dong et al., 2014; Hua et al., 2018), and the potential effects of aerosol redistribution need further discussion. Since this study focuses on the comparison and competition of East and West aerosol forcings, we are specifically interested in how the increasing Asia aerosol forcing affects the North Pacific and how that might be compensated by declining aerosol forcing from North America."*

L 115. I wouldn't call it "full" ACI representation. Obviously, there are many aspects that are missing. I suggest to change it to: "enables simulations of aerosol indirect effects" without the "full".

**Response:**

**Thanks for the suggestion. We deleted "full". We have modified the description as suggested.**

*"The cloud physics scheme allows ice supersaturation and features activation of aerosols to form cloud droplets and ice crystals and thus enables simulations of aerosol indirect effects (Morris and Gettlemen, 2008), which was missing in the model's predecessors."*

L239. If the AOD doesn't change so much far away from the aerosol sources, how come the cloud droplet number concentration change? Have you really checked that the CDNC change?

I think that the cloud fraction changes seen here are dominated by adjustments due to changes in the SST and circulation rather than by aerosol indirect effect.

**Response:**

**Thanks for the valuable comments.**

**We checked the cloud number concentrations (Fig. R1 here; now included in new Fig. 4) and see significant cloud droplet changes over the subtropical Pacific regions due to the aerosol emission over Asia. We also see some cloud droplet changes over the North Pacific region due to aerosol reduction from**

North America. Therefore, we argue that the indirect aerosol effects extend beyond the emission domain and well into the ocean.

[Figure]

**Fig R1 (also included in the new Fig. 4)**

      40-year trend in the percentage Vertical-integrated cloud droplet concentration (top row; CDNUMC; %/decade) and Total cloud fraction (bottom row, CLDTOT, %/decade) in response to FF (left), EastFF (mid), and WestFF (right); (2nd row).

      Although the results above indicate that aerosol indirect forcing (fast response) contributes to the tropical Pacific cloud changes, we also agree with the reviewer that cloud fraction changes are partially driven by the SST and circulation responses (so-called slow response). For example, the eastern subtropical Pacific in the Southern Hemisphere shows cloud fraction changes without much CDNC change.

**Therefore, we modified our discussion on the aerosol-cloud interaction discussion. Now it read:**

*"In line with the zonal asymmetry of AOD_AA trends, simulated solar radiation flux also has significant zonal contrast due to aerosols' direct and indirect climate effects. The first row of Fig. 3 shows the Surface Downward Solar radiation (FSDS), broadly consistent with the patterns of the AOD_AA trend (third row of Fig. 2). Note the opposite colors, though, because a decline in AOD_AA leads to an increase in FSDS. The global surface radiative forcing shows an overall positive trend in response to the decrease in global sulfate emission, but with significant spatial heterogeneity due to the opposite regional emission trends. An increase in FSDS occurs over North America, Europe, and the northern part of Asia, consistent with reducing aerosol forcings over these regions. In contrast, the increasing aerosol emission over east Asia induces a substantial decrease in FSDS. The Net Solar Radiation at the Top-Of-Atmosphere (FSNTOA; the second row of Fig. 3), as the main metric for aerosol forcing, is also consistent with FSDS patterns, but shows more obvious responses over the ocean. Both FSDS and FSNTOA show significant trends over not only the emission domain but also over extended regions into the ocean surface. Due to WestFF, the north Atlantic region shows strong increases in solar radiations, which is consistent with the significant decrease in cloud droplet concentration (CDNUMC, third row of Fig. 3) in response to the WestFF aerosols. However, the cloud fractions (fourth row of Fig. 3) show very weak changes over the north Atlantic, which indicates the critical role of the aerosol first indirect effect over the north Atlantic. The reduction of aerosol emission over North America also leads to smaller cloud droplet concentrations over the North Pacific in the WestFF case, further contributing to a positive radiative forcing. In contrast, in response to EastFF, cloud droplet concentration shows a significant increase over the subtropical Pacific in the Northern Hemisphere, which is consistent with the weak increase in SO4 burden over this region. The larger cloud droplet concentration increases cloud albedo and amplifies the negative*

*radiative forcing. The negative radiative forcing over the subtropical Pacific is evident in the EastFF case, but weaker in the (total) FF case due to the offset by the decreasing aerosols from North America.*

*The North Pacific region, a focused region of this study, shows complex competition of the two emission sources, where WestFF induces a significant decrease in cloud droplet concentration (along with increasing FSNTOA) northward of 30 ºN. In contrast, EastFF leads to an opposite trend at 30 ºN and south. One may expect an increase in FF aerosol over Asia would lead to a negative forcing trend over the North Pacific, as is claimed in previous studies, but actually, the simulated negative trends are confined to lower latitude regions (30 ºN and south). The two sets of regional forcing simulations reveal clearly that the decline of FF aerosol over the WH mid-latitudes induces the positive radiative forcing trend at mid-high latitudes of the North Pacific, producing a weak positive FSNTOA trend. This demonstrated East-West competition is a focal point of our following analysis. In the subsequent sections, we will discuss the possible mechanisms in terms of temperature and circulation changes. "*

**In light of the suggested mechanism of cloud fraction change due to slow response rather than aerosol microphysics, We also added a brief discussion on the possible contribution from the slow response:**

*"The cloud fractions over this region also show an increasing trend in the FF and EastFF, but fail to pass the significance test in the FF case. Surprisingly, the eastern subtropical Pacific in the South Hemisphere also shows significant changes in TOA solar radiation and the cloud fraction, without much aerosol changes. This may possibly be explained by the slow response of sea surface temperature (SST) to the aerosol forcing, where the cloud fraction is affected by the climate adjustment due to SST or circulation changes (Xu and Xie, 2015; Wang et al., 2016; Dong et al., 2019; Kang et al., 2021). The slow responses of cloud fraction to aerosol forcing could also occur near the emission regions where SST changes more significantly; however, as discussed above, the simulated radiation changes over and near the emission regions are highly consistent with the changes in cloud droplet concentrations, indicating a dominant role of indirect aerosol forcing through microphysics perturbation. Here we mainly focused on the overall circulation changes in response to regional aerosol forcings using a fully-coupled climate model, therefore a clear separation of the slow and fast responses of clouds and climate to aerosol forcing is beyond the scope of this study. "*

L241. I might miss something here but as far as I can tell Fig. 3 does not show any significant cloud fraction change over the North-Atlantic in response to FF.

**Response:**

**Thanks for the correction. There is a decrease in cloud droplet numbers rather than the cloud fraction. We have re-written our analysis in this paragraph, including the discussion on cloud droplet numbers (shown in Fig. R1 and added to Fig. 3). Please see our response to the previous comment for the modified discussion.**

L264. Only 2 ensembles are presented in Fig. 4a, not 3.

**Response:**

**Thanks for the correction. We have fixed the issue.**

L 303. Changes in circulation have many implications other than extreme weather.

**Response:**

**Thanks for the suggestion. We modified it to cover more general climate changes:**

*"In this subsection, we discuss the global and regional tropospheric circulation responses due to the evolving anthropogenic aerosol emission, which have a major implication on mid-latitude climate (Xu and Xie, 2015; Mann et al., 2017; Wang et al., 2020)."*

L 306. As was mentioned above, previous studies have also looked at the west-east contrast effect of aerosol on the circulation

**Response:**

**Thanks for the suggestion. We now add related references to the paragraph, copied below:**

*"Previous studies have explored the tropospheric circulation responses to inter-hemispheric (meridional) forcing gradient due to anthropogenic aerosols – more reflecting aerosols over NH compared to SH will lead to an equatorward shift of NH Hadley circulation and NH westerly wind (e.g., Hwang et al., 2013; Hilgenbrink et al., 2018). Meanwhile, recent studies also put effort into how the west-east contrast effects of aerosol induce the circulation changes (Wang et al., 2015; Kang et al., 2021)."*

The paragraph starting at L344. The "rule-of-thumb" explanation presented here is circular and the causality here is not clear. Changes in circulations due to any forcing will derive changes in clouds and humidity, which will derive changes in radiation, which will feed back on the changes in the circulation. Hence, suggesting that the apparent end-result radiation changes (which are also driven by changes in circulation, not just directly by the external forcing) leads to the changes in the circulation is misleading.

The literature is full with explanations and theories about what determines the location of the jet stream. I suggest to change the discussion here to follow the previous knowledge in the field.

**Response:**

  **Thanks for the valuable comments. We agree with the reviewer's point that the circulation changes will induce the radiation changes. However, by introducing the "rule-of-thumb" indicator, we argue that the latitudinal gradient of FSNTOA, a measurable quantity, can be considered as an indicator of the circulation shifts in NH. We do not intend to claim that the circulation changes are simply driven by the FSNTOA gradient. In this revision, we further clarify our point in the manuscript to avoid misleading.**

**As suggested by the reviewer, here we introduce a new analysis on the meridional heat transport (MHT) to provide a deeper causality discussion on the circulation change, specifically for the Hadley Cell shift. Due to the length of the manuscript, we did not include further diagnostics on the NH jet streams. More detailed analyses on jet streams will be presented in a future study.**

**The heavily modified Section 3.3 portion related to MHT now reads:**

*"Previous studies have explored the tropospheric circulation responses to inter-hemispheric (meridional) forcing gradient due to anthropogenic aerosols – more reflecting aerosols over NH compared to SH will lead to an equatorward shift of NH Hadley circulation and NH westerly wind (e.g., Hwang et al., 2013; Hilgenbrink et al., 2018). Meanwhile, recent studies also put effort into how the west-east contrast effects of aerosol induce circulation changes (Wang et al., 2015; Kang et al., 2021). However, from 1980 to 2020, NH anthropogenic aerosol forcing (Sect. 3.1) is highly heterogeneous, with both strong zonal contrasts and subtle latitudinal differences (Fig. 4), further compounding the forcing-response relationship (Shindell and Faluvegi, 2009; Persad and Caldeira, 2018). Next, we analyze the aerosol-induced tropospheric responses in terms of zonal average, both globally and regionally for the EH and WH portions (marked as red boxes in Fig. 4a).*

*Figure 6a–c shows the decadal trend of global Zonal Mean Meridional overturning Stream Function (ZMMSF) in response to FF, EastFF, and WestFF during 1980–2020. The ZMMSF, in response to FF, features a counter-clockwise Hadley Cell anomaly (shown in blue) over the tropics, which indicates a northward shift of the Hadley Cell into NH. The northward shift of Hadley Cell also clearly occurs in response to WestFF, but not to EastFF, indicating that the shift of Hadley Cell is mainly due to the WestFF. The global mean ZMMSF shifts in our results are consistent with previous studies (Xu et al.,*

*2015; Allen and. Ajoku, 2016; Amaya et al., 2018;* Shen *et al., 2018) focusing on the*

*inter-hemispheric forcing gradient. That is, the tropical circulation always tends to move towards a*

*warmer hemisphere with larger positive forcing.*

*To further diagnose why EastFF and WestFF induce distinct changes of the Hadley Cell, we*

*calculated the zonal, column integrated meridional energy transport in response to aerosol forcings,*

*shown in Fig. 7b-d. The Atmospheric Energy Transport (AET) is calculated based on the:*

$$\frac{\partial}{\partial \Phi} F_a = R_{TOA} - Q, \tag{2}$$

*Where $\Phi$ is latitude, $F_a$ (a function of latitude and longitude) is the meridional energy flux, $R_{TOA}$ is*

*the net radiative flux at the top-of-atmosphere (downward positive), and Q is the net downward*

*energy flux at the surface. Q includes shortwave radiation, longwave radiation, sensible heat flux,*

*and latent heat flux. AET is then obtained by integrating the energy flux from south to north:*

$$AET(\Phi) = 2\pi a^2 \int_{-\pi/2}^{\Phi} cos \, \Phi' \, (R_{TOA} - Q) \, d\Phi' \tag{3}$$

*Where a is the Earth radius. The oceanic energy transport (OET) is calculated based on the:The*

*oceanic energy transport (OET) is calculated based on the:*

$$\frac{\partial}{\partial \Phi} F_o = Q \tag{4}$$

*The positive radiative forcing in NH extratropics from WestFF induces a strong negative AET at the*

*equator (Fig. 7d), which leads to the northward shifts of Hadley Cell and ITCZ to balance the*

*interhemispheric difference in radiative forcing. Previous studies demonstrated that cooling NH*

*leads to a southward shift of ITCZ (Broccoli et al., 2006, Kang et al., 2021), and the mechanism is*

*consistent with what we find here. On the other hand, the EastFF introduces a strong negative*

*radiative forcing close to the NH tropics and a weak positive forcing in NH extratropics; as a result,*

*the AET has much smaller trends at all latitudes compared to WestFF (Fig. 7c vs. Fig. 7d). Therefore, the Hadley Cell does not shift significantly in response to EastFF. The AET changes in response to the total FF (Fig. 7b) closely resemble that in response to WestFF, again confirming the dominant role of WestFF in driving the Hadley Cell. OET at the equator in response to all three cases shows a near-zero trend, so it does not contribute much to the shift of Hadley Cell. "*

[Figure]

**Fig R2 (added as the new Figure 7)**

**(a) The 40-year climatology of Northward energy transport (Pwatt) is calculated based on ALL experiments ensemble average. (b–d) The decadal trend of northward energy transport in response to FF, EastFF, and WestFF (Pwatt/decade), which are obtained by subtracting the fixed single forcing experiments from the ALL experiment.**

The dashed blue lines represent the oceanic energy transport; the solid red lines represent the atmospheric energy transport.

In addition, it is not clear here if the focus is on the sub-tropical jet or the eddy-driven jet.

**Response:**

**Thanks for the question. We look at the NH jet that occurs at the subtropical region of around 30 ºN. But in this study we do not intend to separate the subtropical or eddy-driven jet, which are mostly entangled for NH. We further clarify this in the text.**

L371. Obviously, the sub-tropical jet is largely linked to the geostrophic winds. This isn't a surprise.

**Response:**

**Yes, we agree. The subtropical jet largely follows geostrophic winds. By showing the diagnostic geostrophic winds in Figure 7 (now Figure 8), we just want to explain that the shifts of the jet stream are consistent with the simulated SAT gradient.**

**Therefore, we removed the sentence in question:**

*"The sign of SAT gradient supports our previous argument that the jet stream changes are largely linked with Ug. "*.

L446. In the North Atlantic there is a clear signal of the North-Atlantic warming hole (Dagan et al., 2020; Fiedler and Putrasahan 2021;). Worth mentioning here, I think.

**Response:**

**Thanks for the suggestion. We have added more discussion on the Atlantic ocean and set it to be a separate paragraph.**

*"The North Atlantic warming, as many other studies (e.g., Navarro et al., 2017; Qin et al. 2020) pointed out, can be attributed to the reduction of aerosol emission over North America and Europe since the 1980s, which is clearly seen in TOA net energy flux (blue circle in the 2nd row of Fig. 10). A North Atlantic warming hole is also significant in the FF response (Dagan et al., 2020; Fiedler and Putrasahan 2021). Notably, the simulated warming hole is less significant in response to WestFF forcing alone. In response to the EastFF forcing, the high latitudes of the North Atlantic show a warming trend despite insignificant local changes of TOA net energy flux (2nd row of Fig. 10)."*

L430. By "warming forcing" do you mean positive forcing? Similarly, in L 435 "cooling forcing" should be negative forcing.

**Response:**

**Thanks for the suggestion. We have changed all "warming/cooling forcings" to "positive/negative" forcings.**

The explanation around L 435. This is a very descriptive explanation. It is still unclear to me why the differences in the aerosol latitudinal distribution between WH and EH impact the North-Pacific response.

**Response:**

**Thanks for the question. We further discussed the differences between low and high latitude forcings based on the cloud droplet concentrations in Fig. 4 and clarified our descriptions. It now reads:**

*"Let's compare the three sets of responses. The weak cooling over the Pacific warm pool region in response to FF can be explained by the offsetting effects between EastFF and WestFF, where the WestFF-induced warming weakens the strong cooling due to EastFF. Similarly, at the North Pacific region southward of 40 ºN, the extended aerosol cooling effect from East Asia is largely offset by the warming effect due to WestFF in the total FF response. A notable finding is that, at least based on the simulation here, the North Pacific warming northward of 40 ºN is dominated by the positive forcing from WH mid-to-high latitudes, overwhelming the cooling from EH low-to-mid latitudes. The subtle difference in the latitudinal displacement of EH and WH forcings is playing a role here. Indeed, Fig. 4 shows that the EastFF-induced CDNUMC changes are concentrated over low-to-mid latitudes (close to the emission sources and western subtropical Pacific). In contrast, the WestFF-induced CDNUMC changes expand to a larger domain over North Pacific and North Atlantic. The simulation here indicates that mid-to-high latitude SAT is more sensitive to extratropical forcing than forcings originating from a lower latitude. This finding is consistent with previous findings that emission at higher latitudes generates stronger temperature responses (Shindell and Faluvagi, 2009; Persad and Caldeira, 2018). We also examine the BB case (not shown), which has a strong negative forcing over northeastern Asia over 50 ºN, and we find that BB-induced cooling occurs over the entire North Pacific similar to WestFF-induced response. Therefore, we highlight that the latitudinal distribution of aerosol forcing is essential to the North Pacific climate responses. "*

Explanation around L 460. It is not clear to me how you calculated ZMMHT but I assume that it includes only the atmosphere heat transport and not the ocean transport. The surface temperature is strongly controlled also by the ocean heat transport and aerosol forcing could modulate it (Cai et al., 2006; Delworth and Dixon, 2006; Dagan et al., 2020; Menary et al., 2020; Fiedler and Putrasahan 2021; Hassan et al., 2021). This is something that can't be ignored, definitely for the North-Atlantic. This comment about the role of the ocean circulation changes in shaping the SST (or SAT) is true also for many other parts of this paper. It can't be simply ignored.

 **Response:**

**Thanks for the valuable comments. We now included oceanic transport into the discussion in section 3.4 and as the bottom row of Fig. 10 (shown as Fig. R3 below). It now reads:**

*"In addition to the atmospheric ZMMHT, we also show the zonal, column integrated AET response in EH versus WH (bottom row of Fig. 10). The AET results show a strong northward energy transport trend in the WH, while the equatorward energy transport trend in the EH, both of which resemble the ZMMHT trends. Moreover, it is clear that the strong WestFF positive radiative forcing induces the strong poleward energy transport trend in the WH and the AET in response to EastFF is much weaker.*

*Previous studies have also shown that the surface temperature response to aerosol forcing is also strongly modulated by the oceanic energy transport (OET; Cai et al., 2006; Delworth and Dixon, 2006; Dagan et al., 2020; Menary et al., 2020; Fiedler and Putrasahan 2021; Hassan et al., 2021). Here we show that OET (blue lines in Figure 10 bottom row) can have different trends from AET or*

*ZMMHT. In response to the WestFF (positive) forcing, OET increases in WH high latitude and decreases in EH, indicating that poleward heat transport via the ocean is strengthened over the North Atlantic, but is weakened over the North Pacific. The stronger poleward OET in WH, in addition to poleward AET in a larger magnitude, explains why the North Pacific shows a stronger warming trend without local forcing. The EastFF also induces increasing OET in WH and decreasing OET in EH, but with small magnitudes compared with WestFF-induced OET responses, which is similar to AET responses. This further suggests that the WestFF forcing at mid-to-high latitude is the dominant driver of atmospheric and oceanic poleward energy transport in NH."*

[Figure]

*Fig R3 (new Figure 10):*

**1st & 2nd rows: 40-year sea surface air temperature trend (SAT) and net energy flux at TOA (Rtoa). Arctic regions with significant sea ice changes are masked.**

*The blue (red) circles in the 2nd row indicate the major regions with a decline (increase) of aerosol emission.*

*3rd row: 40-year trends of zonal mean meridional heat transport (ZMMHT) in EH and WH. The positive values (solid lines for climatology and red shading for long-term trend) indicate poleward transfer. The arrows indicate the energy transfer responses relative to the climatology.*

*4th row: Similar to Fig. 7 but showing the regional northward energy transport (Pwatt/decade; northward positive) in EH and WH. The dashed blue lines indicate oceanic northward energy transport (OET), while the solid red lines indicate atmospheric northward energy transport (AET).*

L 481-489. This is not a main finding of this paper. The trend of a shift in aerosol forcing from the WH to the EH in recent decades is well known and was discussed in many previous studies.

**Response:**

**Thanks for the suggestion. We have removed well-explained conclusions here, only keeping our major point: the zonal and latitudinal distribution.**

*"(1) The significant zonal contrast in aerosol emission redistribution, and to a lesser extent the latitudinal difference, leads to opposite local radiative forcing that has competing effects on regional climate."*

**Technical comments**

**Response:**

      **Thanks for the corrections. We have carefully re-edited our manuscript and fixed all the issues listed below.**

L69: "et al., 2014". Please correct.

**Changed to "Allen et al., 2014".**

L 90. "Northern Hemisphere, We". Please correct.

**Changed to "Northern Hemisphere. We ".**

L 95. "radiation(Sect. 3.1)"

**Changed to "radiation (Sect. 3.1)".**

L 217. "(1)1"

**Changed to "Eq. (1)".**

L239. "the. U"

**Changed to "the zonal wind speed".**

Fig. 5 some of the y-axis are cut out.

**We replot Fig. 5.**

L 429. "50 North North"

**Changed to "the east coast region northward of 50 ºN. The North Atlantic".**

L430. "induce es strong". This all sentence is not written well.

**Changed to:** *"On the other hand, the WestFF, with positive forcing at WH mid-to-high latitudes (30 ºN–60 ºN; blue oval in Fig. 10), induces a strong warming pattern not only locally at North Atlantic but more so over the North Pacific. ".*

L 468. "othe ver"

**Changed to "over the".**

L527. "More specifically, The"

**Changed to "More specifically, the".**

L 557. You jump from point 3 to 5.

**Changed the order.**

Line 775. "(d–f) Similar to Fig. 6 (d–f)" ? it is also not clear to me if you are presenting here the EH only or also the WH? (all titles say EH).

**Response:**

**Thanks for pointing out the confusion. We show the vertical patterns of EH from the three sets of experiments (FF, EastFF, and WestFF). And we mentioned in the text that the result of global mean and WH are all consistent, thus not shown.**

**We changed the description to:**

*"(d–f) Similar to the bottom row of Fig. 6 but for the U trend in Eastern Hemisphere (EH) from the three sets of experiments. "*

**References**

Cai, W., Bi, D., Church, J., Cowan, T., Dix, M., & Rotstayn, L. (2006). Pan-oceanic response to increasing anthropogenic aerosols: Impacts on the Southern Hemisphere oceanic circulation. Geophysical Research Letters, 33(21). https://doi.org/10.1029/2006gl027513

Chemke, R., & Dagan, G. (2018). The effects of the spatial distribution of direct anthropogenic aerosols radiative forcing on atmospheric circulation. Journal of Climate, 31(17), 7129– 7145.

Dagan, G., Stier, P., & Watson-Parris, D. (2020). Aerosol forcing masks and delays the formation of the North Atlantic warming hole by three decades. Geophysical Research Letters, 47(22), e2020GL090778. https://doi.org/10.1029/2020gl090778

Delworth, T. L., & Dixon, K. W. (2006). Have anthropogenic aerosols delayed a greenhouse gas-induced weakening of the North Atlantic thermohaline circulation? Geophysical Research Letters, 33(2). https://doi.org/10.1029/2005gl024980

Fiedler, S., Stevens, B., & Mauritsen, T. (2017). On the sensitivity of anthropogenic aerosol forcing to model-internal variability and parameterizing a Twomey effect. Journal of Advances in Modeling Earth Systems, 9(9), 1325.

https://doi.org/10.1002/2017MS000932

Fiedler, S., & Putrasahan D. (2021). How Does the North Atlantic SST Pattern Respond to Anthropogenic Aerosols in the 1970s and 2000s? Geophysical Research Letters, https://doi.org/10.1029/2020GL092142

Hassan, T., Allen, R. J., Liu, W., and Randles, C. A. (2021). Anthropogenic aerosol forcing of the Atlantic meridional overturning circulation and the associated mechanisms in CMIP6 models, Atmos. Chem. Phys., 21, 5821–5846, https://doi.org/10.5194/acp-21-5821-2021

Menary, M. B., Robson, J., Allan, R. P., Booth, B. B., Cassou, C., Gastineau, G., et al. (2020). Aerosol-forced AMOC changes in CMIP6 historical simulations. Geophysical Research Letters, 47, e2020GL088166.

Persad, G.G., & Caldeira, K. (2018). Divergent global-scale temperature effects from identical aerosols emitted in different regions. Nat Commun 9, 3289 https://doi.org/10.1038/s41467-018-05838-6

**Response:**

   **Thanks for the references list below, we have cited all references in the revised manuscript.**

---

## Author Comment (AC2)

This study compared the radiative effects of anthropogenic aerosols between western hemisphere and eastern hemisphere in the recent decades and their impacts on the atmospheric circulation. It is an interesting topic, as the interhemispheric contrast of anthropogenic aerosols were often focused in previous studies but the eastern-western hemispheric contrast of anthropogenic aerosols gains much less attention despite this may be the dominant change of anthropogenic aerosols in the NH during the last four decades.

**Response:**

**Thanks for the precious summary of the key point of our study, which emphasizes on the zonal contrast of aerosol forcings.**

The analysis is solid, but some descriptions are not very accurate and some mechanisms are still unclear. Moderate revisions are needed to address these concerns.

**Response:**

**Thanks for the valuable feedback. We further clarified our discussions in the manuscript and have included more dynamical analysis to the manuscript in this round of revision:**

- **The cross-equatorial atmospheric energy transport due to radiative forcing occurring at mid-to-high latitudes leads to the Hadley Cell shifting northward to balance the interhemispheric energy imbalance (Fig 7).**

- **Both atmospheric and oceanic meridional energy transport over the North Atlantic contributes to the remote warming over the North Pacific (Fig 10).**

- **Analysis of the cloud droplet concentration to show the response to aerosol indirect forcings (Fig 4).**

- **The additivity and sensitivity test of the two sets of regional forcings experiments (EastFF and WestFF). (Fig. 3)**

    **Please check the detailed responses to the specific comments below.**

**Detailed comments:**

1. Page 2 L23-26: I don't think you can say the SAT gradient can be used as a predictive metric of NH jet changes, as they are just the same thing based on the thermal wind relationship. Both of them are forced by other factors.

**Response:**

 **Thanks for the correction. By introducing the "rule-of-thumb" indicator, we argue that the latitudinal gradient of FSNTOA, not SAT gradient, a measurable quantity, can be considered as an indicator of the circulation shifts in NH.**

 **By showing the diagnostic geostrophic winds and the corresponding SAT gradient in Figure 7 (now Figure 8), we just want to explain that the jet stream shifts are largely consistent with the simulated SAT gradient.**

 **We modify the text in the abstract as follow:**

*"This leads to a counter-clockwise anomaly of zonal mean stream function over the tropics (i.e. a northward shift of Hadley cell) and stronger equatorward shift of the Northern Hemisphere (NH) jet stream, consistent with the thermal wind argument of surface air temperature (SAT) gradient. Furthermore, the consistent relationship between the jet stream shift and the Top-of-Atmosphere net solar flux (FSNTOA) gradient suggests that the latter can be considered as a rule-of-thumb indicator."*

2. Page 2 L28-29: First, this sentence seems problematic; Second, when you say "dominating role of WH forcing", which aspects do you point to? Based on your results, at least for the jet shift, it is more caused by EH forcing.

**Response:**

**Thanks for the suggestion. We want to emphasize the dominating role of WH forcing in driving NH SST responses. As is pointed out by the reviewer, we realize that our sentence may bring confusion. We modified this paragraph to describe our conclusions more clearly:**

*"Two sets of regional FF simulations (Fix_EastFF1920 and Fix_WestFF1920) are performed to separate the roles of East versus West aerosol forcings, which had clearly opposite trends in the last 40 years. We find that the WH aerosol reduction dominated the simulated warming over NH mid-to-high latitudes. "*

3. L38-40, IPCC AR6 shows it is 1.1C

**Response:**

**Thanks for the correction. We have changed the description from 1C to 1.1C.**

4. L40-42: Dong and Mcphaden (2017) also shows a good example of how the GHGs and internal variability, such as IPO, shapes the global warming at the decadal time scale.

Lu Dong and Michael J McPhaden 2017 Environ. Res. Lett. 12 034011.

**Response:**

**Thanks for the suggestion. We added the reference to the first paragraph of the Introduction section.**

5. L53-55: Similar conclusions have been obtained in earlier studies, such as Salzmann (2016). Salzmann, M. Global warming without global mean precipitation increase? Sci. Adv. 2, e1501572 (2016).

**Response:**

**Thanks for the suggestion, except that Salzmann's study is about global mean precipitation, rather than the precipitation extremes under various intensities as examined in Lin et al. (2018).**

**We added the suggested references to the manuscript.**

6. L44-61: Although there are many differences between GHGs and aerosols as you mentioned here, some other studies found some climate impacts caused by GHG and aerosols can be very similar. For example, Xie et al. (2013) showed the SST and ocean precipitation response patterns are very similar in both GHGs and aerosols. Recently,

Song et al. (2021) also shows both GHG and aerosols modulate the seasonal delay of tropical rainfall in a similar way, i.e., by modulating the atmospheric column humidity. This similarity between the two forcings should also be mentioned.

Xie, SP., Lu, B. & Xiang, B. Similar spatial patterns of climate responses to aerosol and greenhouse gas changes. Nature Geosci 6, 828–832 (2013).

Song, F., Leung, L.R., Lu, J. et al. Emergence of seasonal delay of tropical rainfall during 1979– 2019. Nat. Clim. Chang. 11, 605–612 (2021).

**Response:**

**Thanks for the suggestion. We add such discussion on the similar forced responses to the introduction.**

*"However, despite the subtle differences between GHGs and aerosols, other studies found similar climate responses to GHGs and aerosols. Xie et al. (2014) found that the regional ocean temperature and precipitation in response to GHGs and aerosols are similar, suggesting the importance of the spatial distribution of radiative changes. Song et al. (2021) show that increasing GHGs and decreasing aerosols in the recent decades both delay rainfall by inducing a moister atmosphere. Both the differences and similarities between GHGs- and aerosol-induced climate responses indicate the complexity and importance of the temporal and spatial distribution of AA forcings. "*

7. L59-61: As mentioned above, Song et al. (2021) found the recent decreases of aerosols, combined with the increased GHGs, contribute significantly to the seasonal delay of tropical rainfall. As the decreased aerosol and increased GHG will continue in the future, the seasonal delay of tropical rainfall is expected to amplify in the future.

**Response:**

**Thanks for the suggestion. We have added the following to the text:**

*"The FF-related aerosols are projected to further decrease in future decades (Andreae et al., 2005; Zheng et al., 2020), even for Asian regions, with more strict air quality measures in developing nations. The future decline of FF aerosol will lead to further unmasking and warming in addition to GHG-induced global warming (Xu et al., 2015; Wang et al., 2018; Lelieveld et al., 2019; Allen et al., 2020) and have consequences for heat extremes (Zhao et al., 2019; Xu et al., 2020) and humidity and precipitation (Song et al., 2021)."*

8. L69: for the first reference: Who?

**Response:**

**Thanks for pointing out the editing issue. Should be "Allen et al., 2014"**

9. L78: tropics->tropical

**Response:**

**Thanks for the correction. We have fixed this issue.**

10. L166-167: Fix_FF1920 have 20 members, but here Fix_EastFF1920 and Fix_WestFF1920 only contain 10 members, is there any sensitivity of the results to the member numbers? For example, if you also only use 10 members of Fix_FF1920, could you obtain the similar results?

**Response:**

**Thanks for the suggestion. We conducted sensitivity tests to see if ten ensemble members are sufficient enough to smooth out the randomly generated internal variabilities in the model. We calculated the Tripole Index for the Interdecadal Pacific Oscillation (TPI; Henley et al., 2015) based on each individual simulation and the multi-member-mean (MMM) results for each experiment, which is shown in Fig. R1.**

**Although we still see some small variations in the MMM TPI index of Fix_EastFF1920 and FixWestFF1920, the majority of the internal variability is successfully smoothed out. There is no doubt that the more ensemble members exit, the less model-generated internal variability will exist in MMM results, but considering that Fix_FF1920 and the two regional experiments show the similar magnitude of variation, we believe that 10 members are acceptable for the purpose of separating externally driven responses and the randomly distributed variability.**

Henley, B.J., Gergis, J., Karoly, D.J., Power, S.B., Kennedy, J., Folland, C.K., (2015). A Tripole Index for the Interdecadal Pacific Oscillation. Clim. Dyn. 45 (11–12), 3077–3090, http://dx.doi.org/10.1007/s00382-015-2525-1.

[Figure]

**Fig R1:**

Model-generated TPI index from (a) ALL, (b) Fix_EastFF1920, (c) Fix_FF1920, and (d) Fix_WestFF1920. The grey lines represent each ensemble member; the black curve represents the TPI index calculated from the ensemble average. Note that ALL has 40 members; Fix_FF1920 has 20 members; Fix_EastFF1920 and Fix_WestFF1920 have 10 members, respectively.

Another relevant question is that you should also show whether the trend of many variables you focused here in the Fix_FF1920 is roughly the sum of Fix_EastFF1920 and Fix_WestFF1920.

**Response:**

Thanks for the suggestion. We conducted the additivity test as suggested on the burden of sulfate aerosol and surface temperature (Fig. R2).

For the column burden of sulfate, the results in response to FF resemble the sum of EastFF and WestFF (SUM).

For the surface temperature (TS), FF-induced TS trends are also similar to the sum of EastFF and WestFF, except for the central Pacific and part of the Arctic region. The possible reason for the slightly larger warming in SUM may be related to the residue of internal variability (as shown in Figure R1 above) due to the limited ensemble sizes, the responses to the unfixed regions (such as the Arabian Peninsula and Africa), or the nonlinearity issues due to model subtraction. Overall, we argue that the sum of the two regional experiments (EastFF and WestFF) show very similar responses compared with FF responses and thus are capable of separating the East versus West aerosol forcings.

We add the following discussion to the Method section:

*"An additivity test is conducted to evaluate whether the summation of EastFF and WestFF can roughly reproduce FF. The SO4 column burden (BURDENSO4) and surface temperature (TS) in response to FF and EastFF + WestFF (SUM hereafter) are shown in Fig. 3. The FF-induced SO4 column burden resembles the sum of the SO4 burden from SUM. The TS responses are also very similar between FF and SUM, except for the central Pacific and part of the Arctic region. The warmer patterns over the central Pacific in SUM compared to FF is possibly related to the TS responses to remote forcings beyond the two regions in consideration here (e.g., Arabian Peninsula, South America, and Africa), the residues effects of internal variability even after ensemble average due to limited ensemble sizes. Overall, the sum of two sets of regional fixed*

*single forcing experiments well represent the major patterns of FF aerosol induced response, and*

*thus the two new sets of simulations here are capable of separating the East versus West aerosol*

*forcings."*

[Figure]

**Fig R2 (the new Fig. 3):**

**Left column: the 40-year trend of (top) Sulfate column burden (kg/km2/decade), and (bottom) surface air temperature in response to FF.**

**Right column: So as to the left column for the summation of EastFF and WestFF.**

11. L201-202: We know the decreased trend of FF-related aerosols in the North America and Europe is due to the clean air acts and increased trend of FF-related aerosol in the India and China is due to the economic development, but what's the reason of the stronger increasing trend of BB-related POM over the northeastern Asia?

**Response:**

Thanks for the question. Based on Deser et al. (2020) and Lamarque et al. (2010), the BB-related POM over northeastern Asia is due to the increasing biomass combustion from forest fires.

Deser, C., et al. (2020). Isolating the Evolving Contributions of Anthropogenic Aerosols and Greenhouse Gases: A New CESM1 Large Ensemble Community Resource, Journal of Climate, 33(18), 7835-7858.

Lamarque, J.-F., et al. (2010). Historical (1850–2000) gridded anthropogenic and biomass burning emissions of reactive gases and aerosols: methodology and application, Atmos. Chem. Phys., 10, 7017–7039, https://doi.org/10.5194/acp-10-7017-2010

12. L217: removing "1"

**Response:**

Thanks for the correction. We have fixed this issue.

13. L241: and FF-> FF and.

**Response:**

Thanks for the correction. We have fixed this issue.

14. L238-240: Could you explain a little bit more about how the indirect effects of aerosols (i.e., cloud droplets number and cloud lifetimes are enhanced) could expand the affected regions? Do you mean the cloud formed in the emission region can be transported to other places?

**Response:**

**Thanks for the question. Reviewer #1 also touched on this issue.**

**We include the cloud number concentrations in Fig. 3 (the new Fig. 4) and see significant cloud droplet changes over the subtropical Pacific regions due to the aerosol emission over Asia. We also see some cloud droplet changes over the North Pacific region due to aerosol reduction from North America. Therefore, we argue that the indirect aerosol effects extend beyond the emission domain and well into the ocean.**

**Although the results above indicate that aerosol indirect forcing (fast response) contributes to the tropical Pacific cloud changes, we also agree with reviewer #1 that cloud fraction changes are partially driven by the SST and circulation responses (so-called slow response). For example, the eastern subtropical Pacific in the Southern Hemisphere shows cloud fraction changes without much CDNC change.**

15. L244: should be decrease of CLDTOT rather than increase in response to WestFF based on Fig. 3?

**Response:**

Thanks for the question, but actually, there is an increase of CLDTOT over the Pacific warm pool region in Fig 3

[Figure]

**Fig. R3 (the new Fig. 4)**

**The 40-year trend in cloud droplet number concentration (CDNUMC) and total cloud fraction (CLDTOT) in response to FF, EastFF, and WestFF.**

16. Figure 3. You mentioned that regions passing the 95% significance is dotted, but I didn't see any dots there. You may also need to do the significance test in Fig. 2 and many other figures.

**Response:**

Thanks for pointing out the editing mistake. We now added the significance test to the horizontal pattern figures (the new Fig. 4 and Fig. 10).

For the vertical pattern figures, we have masked insignificant regions to be white.

17. Fig. 5: What did you do when you say "smoothed in 30 degrees of latitudinal range"

**Response:**

**We mean we smooth the latitudinal profile using a moving average with a 30DegLat window. We modified the sentence to clarify our method:**

*"The trend and gradient lines are smoothed using the moving average method (with 30 degrees of latitudinal range sampling window)."*

18. would you like to mention how the increased FSNTOA gradient drives the equatorward shift of the NH jet stream?

**Response:**

**Thanks for the suggestion. By introducing the "rule-of-thumb" indicator, we argue that the latitudinal gradient of FSNTOA, a measurable quantity, can be considered as an indicator of the circulation shifts in NH. We do not intend to claim that the circulation changes are simply driven by the FSNTOA gradient. We further clarify our point in the manuscript to avoid misleading.**

**However, since reviewer #1 also suggested adding some dynamic analysis, we now include a new subsection discussing the relationship between the shift of circulation and the cross-equatorial atmospheric energy transport (AET0), specifically for the Hadley Cell shift. Due to the length of the manuscript, we did not include further diagnostics on the NH jet streams. More detailed analyses on jet streams will be presented in a future study, as mentioned in the Conclusion.**

**The heavily modified Section 3.3 portion related to meridional energy transport now reads:**

*"Previous studies have explored the tropospheric circulation responses to inter-hemispheric (meridional) forcing gradient due to anthropogenic aerosols – more reflecting aerosols over NH compared to SH will lead to an equatorward shift of NH Hadley circulation and NH westerly wind (e.g., Hwang et al., 2013; Hilgenbrink et al., 2018). Meanwhile, recent studies also put effort into how the west-east contrast effects of aerosol induce circulation changes (Wang et al., 2015; Kang et al., 2021). However, from 1980 to 2020, NH anthropogenic aerosol forcing (Sect. 3.1) is highly heterogeneous, with both strong zonal contrasts and subtle latitudinal differences (Fig. 4), further compounding the forcing-response relationship (Shindell and Faluvegi, 2009; Persad and Caldeira, 2018). Next, we analyze the aerosol-induced tropospheric responses in terms of zonal average, both globally and regionally, for the EH and WH portions (marked as red boxes in Fig. 4a).*

*Figure 6a–c shows the decadal trend of global Zonal Mean Meridional overturning Stream Function (ZMMSF) in response to FF, EastFF, and WestFF during 1980–2020. The ZMMSF, in response to FF, features a counter-clockwise Hadley Cell anomaly (shown in blue) over the tropics, which indicates a northward shift of the Hadley Cell into NH. The northward shift of Hadley Cell also clearly occurs in response to WestFF, but not to EastFF, indicating that the shift of Hadley Cell is mainly due to the WestFF. The global mean ZMMSF shifts in our results are consistent with previous studies (Xu et al., 2015; Allen and. Ajoku, 2016; Amaya et al., 2018;* Shen *et al., 2018) focusing on the inter-hemispheric forcing gradient. That is, the tropical circulation always tends to move towards a warmer hemisphere with larger positive forcing.*

*To further diagnose why EastFF and WestFF induce distinct changes of the Hadley Cell, we calculated the zonal, column integrated meridional energy transport in response to aerosol forcings, shown in Fig. 7b-d. The Atmospheric Energy Transport (AET) is calculated based on the:*

$$\frac{\partial}{\partial \Phi} F_a = R_{TOA} - Q, \tag{2}$$

*Where $\Phi$ is latitude, $F_a$ (a function of latitude and longitude) is the meridional energy flux, $R_{TOA}$ is the net radiative flux at the top-of-atmosphere (downward positive), and Q is the net downward energy flux at the surface. Q includes shortwave radiation, longwave radiation, sensible heat flux, and latent heat flux. AET is then obtained by integrating the energy flux from south to north:*

$$AET(\Phi) = 2\pi a^2 \int_{-\pi/2}^{\Phi} cos\, \Phi' (R_{TOA} - Q)\, d\Phi' \tag{3}$$

*Where a is the Earth radius. The oceanic energy transport (OET) is calculated based on the:The oceanic energy transport (OET) is calculated based on the:*

$$\frac{\partial}{\partial \Phi} F_o = Q \tag{4}$$

*The positive radiative forcing in NH extratropics from WestFF induces a strong negative AET at the equator (Fig. 7d), which leads to the northward shifts of Hadley Cell and ITCZ to balance the interhemispheric difference in radiative forcing. Previous studies demonstrated that cooling NH leads to a southward shift of ITCZ (Broccoli et al., 2006, Kang et al., 2021), and the mechanism is consistent with what we find here. On the other hand, the EastFF introduces a strong negative radiative forcing close to the NH tropics and a weak positive forcing in NH extratropics; as a result, the AET has much smaller trends at all latitudes compared to WestFF (Fig. 7c vs. Fig. 7d). Therefore, the Hadley Cell does not shift significantly in response to EastFF. The AET changes in response to the total FF (Fig. 7b) closely resemble that in response to WestFF, again confirming the dominant*

*role of WestFF in driving the Hadley Cell. OET at the equator in response to all three cases shows a*

*near-zero trend, so it does not contribute much to the shift of Hadley Cell. "*

[Figure]

**Fig R4 (added as the new Figure 7)**

**(a) The 40-year climatology of Northward energy transport (Pwatt) is calculated based on ALL experiments ensemble average. (b–d) The decadal trend of northward energy transport in response to FF, EastFF, and WestFF (Pwatt/decade), which are obtained by subtracting the fixed single forcing experiments from the ALL experiment.**

**The dashed blue lines represent the oceanic energy transport; the solid red lines represent the atmospheric energy transport.**

19. L371: below 35N? seems problematic. Suggest changing to southward of 35N

**Response:**

    **Thanks for the suggestion. We now change all "below X ºN" to "southward of X ºN", and all "above Xº N" to "northward of X ºN".**

20. Fig. 6: Here, why do you only focus on the EH, rather than Global in previous figures? Could you explain it a little bit?

**Response:**

    **Thanks for the question. We include Fig. 6 (now Fig. 8) to compare geostrophic wind theory (upper row) and the shift of NH jet stream (lower row). Since we already show the global jet stream in Fig. 5 (now Fig. 6), we show EH jet and geostrophic wind in Fig. 6 as an example to avoid duplicated panels. Actually, the consistency between geostrophic wind and jet stream holds true in all cases of ALL, EH, and WH. We mentioned this in the text:**

*"The derived Ug patterns always resemble the simulated U pattern in EH (Fig. 8d–f), WH, and Global (not shown), revealing the strong correlation between tropospheric circulation changes and the tropospheric temperature changes (and thus the geopotential height changes)."*

21. L394: sometimes using FSDS, but in other cases, you use FSNTOA. Please justify your choices.

**Response:**

Thanks for the suggestion. This is due to the similar rationale of Comment/Response 20 above. We previously used FSDS for the shift of Hadley Cell (previous Fig. 5; has been removed in the new Fig. 6 now) and FSNTOA for the jet stream (the new Figure 6). Both FSDS and FSNTOA are treated as the "rule-of-thumb" indicators of the circulation shift. In Fig. 5 (the new Fig. 6), we compare these two and find that these two gradients are pretty similar.

As suggested by the reviewer, and considering the similarity between FSNTOA and FSDS gradient, we now use FSNTOA as the rule-of-thumb indicator throughout the manuscript.

22. L418: references are needed here.

**Response:**

Thanks for the suggestion. We have added references to the text.

*"which is mentioned by previous studies that demonstrate the north Pacific cooling due to Asia aerosol emissions (Dong et al., 2014; Takahashi and Watanabe, 2016; Smith et al., 2016)."*

Takahashi C, Watanabe M (2016) Pacific trade winds accelerated by aerosol forcing over the past two decades. Nat Clim Chang 6:. https://doi.org/10.1038/nclimate2996

Dong L, Zhou T, Chen X (2014) Changes of Pacific decadal variability in the twentieth century driven by internal variability, greenhouse gases, and aerosols. Geophys Res Lett. https://doi.org/10.1002/2014GL062269

Smith DM, Booth BBB, Dunstone NJ, et al. (2016) Role of volcanic and anthropogenic aerosols in the recent global surface warming slowdown. Nat Clim Chang 6:936–940. https://doi.org/10.1038/nclimate3058

23. L430: induce es->induces a

**Response:**

Thanks for the correction. We have fixed this issue.

24. L468: other ver?

**Response:**

Thanks for the correction. It should be "over". We modified this sentence and fixed the error.

---

## Author Response (AR2)

Review of:

Anthropogenic Aerosol effects on Tropospheric Circulation and Sea Surface Temperature (1980-2020): Separating the role of Zonally Asymmetric Forcings

By Diao et al.

I would like to thank the authors for replying to my comments. I stand by my previous opinion that the new simulations presented here are well executed set of numerical experiments, which presents a valuable contribution to the climate community. In my opinion, the paper now is in a much better shape and should be ready for publication after a minor revision. I have a few comments and suggestions for the authors:

**Response:**
**We really appreciate the reviewer's careful reading and valuable suggestions, which helped us improve our paper's quality. Furthermore, We realized that some parts of the manuscript were hard to read. Therefore, we have updated those parts (e.g., abstract, introduction, and summary parts) to improve the readability and clarify our arguments.**
**Please check our response to the major comments below.**

• I still would like to encourage the authors to conduct a carful re-writing and editing of the paper. I am not an expert myself, but the level of English of this paper is low in many parts.

**Response:**
**Thanks very much for pointing out the language issues. We carefully re-edited our manuscript and corrected many editorial issues.**

• L25 (and L221, L318, L392, L424, L437…): I suggest to stick to the IPCC definition of radiative forcing. The decline in aerosol concentration by itself does not cause a positive radiative forcing (comparers to pre-industrial conditions) but rather just a decrease in the value of the negative radiative forcing.

**Response:**
**Thanks for the suggestion. We agree with the reviewer that decline in aerosol causes positive radiation anomaly instead of positive radiative forcing. In the abstract (L25), we changed the**

**description to:** "*weakening negative radiative forcing over WH mid-to-high latitudes and enhancing negative radiative forcing over EH at lower latitudes*"

**We also added a clarification on line 221. For other lines, we now use the "positive radiative forcing anomaly" or "positive trend of radiative forcing".**

**Line 221 now reads:**
"

*In Fix_EastFF1920, except for the increasing negative forcing in the lower latitudes of East Asia, Siberia shows a slight weakening of the negative forcing (positive anomaly of radiative forcing) due to the extension of WH aerosol reduction.*
"

• L75: I suggest to remove the "subtle". I think that the differences between GHG and aerosol forcing is not subtle.

**Response:**

**Thanks for the suggestion. We have removed the "subtle" in line 75 and many other places as suggested.**

• L78: I don't understand how the fact that it was found that the response to GHG forcing (well mixed) is similar to aerosol forcing (heterogenous) "suggesting the importance of the spatial distribution of radiative changes"? If anything, it suggest that the spatial distribution is not that important.

**Response:**
**Thanks for the comment. We have rewritten the paragraph to avoid this issue. It now reads:**
"

*However, despite the differences between GHGs and aerosols, other studies found similar climate responses to GHGs and aerosols. Xie et al. (2013) found that the 20th century regional temperature and precipitation are similar in response to GHGs and the more spatially heterogeneous aerosol forcing.*
"

• L296: why do you call these differences "subtle"? I would remove it.

**Response:**

**Thanks for the correction. We mis-used the "subtle" word. We have changed the "subtle zonal differences" to "significant zonal differences".**

• L313: cloud droplet number concentration is usually denoted by CDNC. I suggest to stick to that and not CDNUMC (I think that it will be easier for most people to follow).

**Response:**

**Thanks for the suggestion. Previously we used the CESM1 model-output name "CDNUMC" to represent the cloud droplet number concentration. Now we have replaced it with "CDNC" in both text and Fig. 4.**

• L316: please add a reference for the aerosol first indirect effect.

**Response:**

**Thanks for the suggestion. We have added the reference, and it now reads:**
**"**

*In response to WestFF, the north Atlantic region shows strong increases in solar radiations, which is consistent with the significant decrease in cloud droplet number concentration (CDNC, third row of Fig. 3). However, the cloud fractions (fourth row of Fig. 3) show very weak changes over the north Atlantic, which indicates the critical role of the aerosol first indirect effect over the north Atlantic (Penner et al. 2001).*
**"**

**Penner, J. E., Andreae, M. O., Annegarn, H., Barrie, L., Feichter, J., Hegg, D., and Pitari, G.: Aerosols, their direct and indirect effects, In Climate Change 2001: The Scientific Basis. Contribution of Working Group I to the Third Assessment Report of the Intergovernmental Panel on Climate Change, Cambridge University Press, 289-348, 2001.**

• L393: again, the differences here are also not "subtle".

**Response:**

**Thanks for the correction. We have removed the "subtle" here.**

• L410: again, "subtle" in a wrong place. You miss-use and over-use this word. Maybe you meant to say substantial instead?

**Response:**

**Thanks for pointing out the problem. By saying "with both strong zonal contrasts and subtle latitudinal differences", we meant that the latitudinal difference is less obvious than the zonal difference. We realized that the "subtle difference" is very misleading, so we have replaced it with "with both zonal and latitudinal contrasts".**

• The paragraph around L560 needs to be re-write. The English level is bad.

**Response:**

**Thanks for the suggestion. We have rewritten the whole paragraph, and it now reads:**
"

*Over the tropical Pacific region, EastFF induces an El Niño-like SAT pattern with symmetric warming trends (does not pass the 95% significance test, though). The EastFF-induced El Niño-like pattern contradicts some previous studies arguing that Asian aerosols lead to a La Niña-like pattern (Kaufmann et al., 2011; Smith et al., 2016; Kang et al., 2021). On the other hand, WestFF induces an asymmetric SAT pattern over the tropical Pacific, with warming in the north and cooling in the south. The distinct tropical Pacific SAT responses due to EastFF and WestFF may also contribute to the Pacific decadal to multi-decadal variability (PDV) in amplitude and spatial pattern. The question about whether and how regional aerosol forcings affect PDV needs further investigation.*
"

• L572: another miss-use of "subtle".

**Response:**

**Thanks for the comment. We have rewritten the sentence, and it now reads:**
"
*The latitudinal difference between EH and WH forcing distribution plays an important role here.*
"

• L60: missing "and", I think.

**Response:**

**Thanks for the comment. We have added an "and" to the sentence.**

**Response:**

**Thanks for the comment. We have fixed this.**

**Response:**

**Thanks for the comment. We have removed the "that" word.**

**Response:**

**Thanks for the comment. There should be an "or". We have rewritten the sentences, and it now reads:**

"

*The surface temperature responses are also very similar between FF and SUM, except for the central Pacific and part of the Arctic region. The warm bias over the central Pacific in SUM is possibly associated with forcings outside the two focused regions (EH box and WH box in Fig. 2), or it is due to the residual effect of internal variability even after ensemble average due to limited ensemble sizes.*

"

• Around L287: stick to either AODAA (underline AA) or AOD_AA

**Response:**

**Thanks for the comment. It is an editing error. We use AOD_AA throughout the manuscript.**

• L298: please correct "climate responses in response"

**Response:**

**Thanks for the comment. We have changed it to: "climate responses to aerosol increase over EH and aerosol reduction over WH."**

• L545: "which is mentioned by previous studies that the north Pacific cooling due to Asia aerosol emissions" – the English here needs to be improved.

**Response:**
**Thanks for the suggestion. We have rewritten the related sentences for better readability. It now reads:**
"

*EastFF induces significant cooling over the western part of the North Pacific at low-to-mid latitudes, which is consistent with previous studies (Dong et al., 2014; Takahashi and Watanabe, 2016; Smith et al., 2016). In contrast, WestFF, with positive forcing anomaly at mid-to-high latitudes (30 ºN–60 ºN; blue oval in Fig. 10), induces large-scale warming locally at North Atlantic and even stronger warming over the entire North Pacific. Thus, the WestFF-induced warming over the North Pacific largely offsets the EastFF-induced cooling in the FF case.*
"

• L636: please remove the: "

**Response:**
**Thanks for the comment. We have fixed this issue.**